# Dbf4-dependent kinase promotes cell cycle controlled resection of DNA double-strand breaks and repair by homologous recombination

Lorenzo Galanti[1,2,3,4], Martina Peritore [2,3,4,10], Robert Gnügge[5], Elda Cannavo[6], Johannes Heipke [1,2,4], Maria Dilia Palumbieri[3,4,11], Barbara Steigenberger[7], Lorraine S. Symington [5,8], Petr Cejka[6,9] & Boris Pfander [1,2,3,4] ✉

DNA double-strand breaks (DSBs) can be repaired by several pathways. In eukaryotes, DSB repair pathway choice occurs at the level of DNA end resection and is controlled by the cell cycle. Upon cell cycle-dependent activation, cyclin-dependent kinases (CDKs) phosphorylate resection proteins and thereby stimulate end resection and repair by homologous recombination (HR). However, inability of CDK phospho-mimetic mutants to bypass this cell cycle regulation, suggests that additional cell cycle regulators may be important. Here, we identify Dbf4-dependent kinase (DDK) as a second major cell cycle regulator of DNA end resection. Using inducible genetic and chemical inhibition of DDK in budding yeast and human cells, we show that end resection and HR require activation by DDK. Mechanistically, DDK phosphorylates at least two resection nucleases in budding yeast: the Mre11 activator Sae2, which promotes resection initiation, as well as the Dna2 nuclease, which promotes resection elongation. Notably, synthetic activation of DDK allows limited resection and HR in G1 cells, suggesting that DDK is a key component of DSB repair pathway selection.

The cell cycle is controlled by a series of transcriptional and post-translational mechanisms that lead to activation and inactivation of cyclin-dependent kinase (Cdk1, Cdc28 in budding yeast, referred to as CDK hereafter), which is often considered as the master regulator of the cell cycle[1]. Cell cycle-dependent signaling, which adjusts cells to the specific requirements of the respective cell cycle phase, involves not only CDK, but several additional cell cycle kinases. Of these, Dbf4-dependent kinase (DDK) has been shown to act in conjunction with

[1]Cell Biology, Dortmund Life Science Center (DOLCE), TU Dortmund University, Faculty of Chemistry and Chemical Biology, Dortmund, Germany. [2]Research Group DNA Replication and Genome Integrity, Max Planck Institute of Biochemistry, Martinsried, Germany. [3]Genome Maintenance Mechanisms in Health and Disease, Institute of Aerospace Medicine, German Aerospace Center (DLR), Cologne, Germany. [4]Institute for Genome Stability in Aging and Disease, University of Cologne, Medical Faculty, CECAD Research Center, Cologne, Germany. [5]Department of Microbiology & Immunology, Columbia University Irving Medical Center, New York, NY, USA. [6]Institute for Research in Biomedicine, Faculty of Biomedical Sciences, Università della Svizzera Italiana (USI), Bellinzona, Switzerland. [7]Mass Spectrometry Core Facility, Max Planck Institute of Biochemistry, Martinsried, Germany. [8]Department of Genetics & Development, Columbia University Irving Medical Center, New York, NY, USA. [9]Department of Biology, Institute of Biochemistry, Eidgenössische Technische Hochschule (ETH), Zürich, Switzerland. [10]Present address: DSB Repair Metabolism Laboratory, The Francis Crick Institute, London, UK. [11]Present address: Research Group of Proteomics and ADP-Ribosylation Signaling, Max Planck Institute for Biology of Ageing, Cologne, Germany. ✉e-mail: boris.pfander@tu-dortmund.de

CDK and - being active from early S-phase to M-phase - shares a similar activation profile[2]. Processes that are under dual control of CDK and DDK include the initiation of DNA replication[3–7], the block to over-replication[8], the initiation of meiotic recombination[9,10], chromosome segregation[11–14] and the resolution of recombination intermediates[15].

DNA double-strand breaks (DSBs) are a severe form of DNA damage, but eukaryotes have evolved a number of repair pathways[16]. DSB repair pathway choice, i.e. the cellular decision of which mechanism to utilize for DSB repair, is controlled at the step of DNA end resection[17]. Resection is a nucleolytic process that degrades the 5' end of DNA ends and generates 3' single-stranded DNA (ssDNA) overhangs, which are critical for repair by homologous recombination (HR)[18]. Limited resection can also expose microhomologies on both ends of the DSB and thereby promote intrinsically error-prone repair by microhomology mediated end-joining (MMEJ)[19,20]. In contrast, resection destroys the DNA substrate for non-homologous end-joining (NHEJ)[21,22]. Mechanistically, resection can be subdivided into resection initiation (short-range resection), which is catalyzed by the Mre11 complex (Mre11-Rad50-Xrs2 (MRX) in budding yeast, MRE11-RAD50-NBS1 (MRN) in human)[23–27], and resection elongation (long-range resection), which is catalyzed by one of two nucleases, Exo1 or Dna2, the latter in conjunction with the STR/BTR complex (Sgs1-Top3-Rmi1 in budding yeast, BLM-TOP3A-RMI1/2 in human cells)[24,28–32]. Resection initiation involves an endonucleolytic cleavage in proximity of the DSB and 3'–5' exonucleolytic resection towards the DSB, which helps to remove protein blocks from DNA ends[27,32–37]. The endonucleolytic clipping also serves as entry point for exonucleolytic resection elongation, which results in 3' overhangs that can stretch for up to several thousand nucleotides[28,29]. In order to proceed through a chromatinized DNA substrate, DNA end resection involves additional factors and is coupled to nucleosome eviction[38].

The key cellular determinant of both resection and HR is the cell cycle. Throughout eukaryotes, resection initiation and elongation are upregulated from S-phase to M-phase[39–42]. Thereby, activation of resection is coincident with activation of DNA replication, a correlation which is rationalized by HR being upregulated when genetic information from the sister chromatid can be used as a template for DSB repair by HR[39–42]. The cell cycle-dependent activation of resection depends on CDK and affects both resection initiation and elongation[43,44]. In the case of resection initiation in budding yeast, CDK has been shown to phosphorylate Sae2[45], which in turn activates the endonucleolytic activity of the Mre11 complex, even though only partially[25,46]. CDK phosphorylation of Sae2 is conserved to human CtIP[47]. In the case of resection elongation, CDK has been shown to target Dna2 in budding yeast[48] and EXO1 in human[49]. Moreover, the resection-promoting nucleosome remodeler Fun30 (Fun30 in budding yeast, SMARCAD1 in human) has been shown to be phosphorylated by CDK in both budding yeast and human systems[50,51].

CDK phosphorylation-mimetic mutations have been introduced into Sae2 and Dna2. However, these mutations do not fully activate DNA end resection and do not bypass cell cycle regulation[45,46,48,51]. Moreover, CDK-phosphorylated Sae2 does not stimulate the Mre11 complex to the same extent as Sae2 purified from cells in phosphorylated form[46]. Therefore, CDK phosphorylation seems not to be sufficient for the timed activation of DNA end resection during the cell cycle, and additional cell cycle-regulated mechanisms remain to be uncovered.

In this study, we showed that – analogously to DNA replication initiation – also resection initiation and elongation are under dual cell cycle control by both CDK and DDK. We find DDK to be required for resection and HR in budding yeast as well as for resection in human cells. Using phospho-proteomics and directed screening of resection factors in yeast, we find a wide-spread involvement of DDK in phosphorylation of DNA repair proteins. Mechanistically, we show that DDK is required for efficient DNA end resection via phosphorylation of both

Sae2 and Dna2. Corresponding phosphorylation-deficient mutants are repair-deficient, suggesting that dual phosphorylation of both proteins is required for cell cycle-dependent activation of resection. Lastly, engineered activation of DDK in G1-phase in budding yeast allows limited activation of resection and HR, indicating a key function of DDK in resection regulation and broadening its role as regulator of genome stability.

## Results

### DDK phosphorylates HR proteins and is required for HR

To investigate if DDK is important for the cellular response to DSBs, we tested chronic exposure to the topoisomerase I inhibitor camptothecin (CPT), known to induce DSBs in cycling cells. Given that DDK is essential, we used the *bob1-1* (*MCM5-P83L*) background, which bypasses DDK's essential role in DNA replication initiation and allows to deplete cells of the kinase *CDC7*, or its regulatory subunit *DBF4*[52]. Deletion of *CDC7* or *DBF4* led to hypersensitivity to CPT (Fig. 1a), consistent with a putative role in the DSB response. To specifically test for an involvement in HR, we used a gene conversion assay, where an HO-induced DSB at Chr. IV, 491 kb can be repaired by HR using an ectopic donor sequence integrated in the same chromosome (Chr. IV, 795 kb). Due to heterology in the donor sequence, repair can be quantitatively measured using qPCR[53] (Fig. 1b). We were concerned about indirect effects caused by aberrant cell cycle progression of cells lacking a major cell cycle kinase such as DDK. Therefore, DDK depletion experiments were carried out with cells arrested in M-phase using nocodazole (Supplementary Fig. 1a). We observed the accumulation of the recombination repair product after DSB induction in *bob1-1* control cells carrying the donor template, but not when *CDC7* was deleted (Fig. 1c), suggesting that DDK is required for HR. Consistently, we observed similar results when we measured HR-dependent cell survival, after induction of an HO-induced DSB (Supplementary Fig. 1b).

To identify DDK phosphorylation substrates, we compared wild type (WT), *bob1-1,* and *bob1-1 dbf4Δ* cells arrested in M-phase by phospho-proteomics (Supplementary Fig. 1c, d). Our analysis revealed a number of phospho-sites, which were reproducibly lost in *bob1-1 dbf4Δ*, but not in control cells (WT and *bob1-1*) (Fig. 1d). In previous studies, DDK was suggested to preferentially phosphorylate serine (S) or threonine (T) residues that are followed by a negatively charged amino acid (D, E, phospho-S/T in +1 position)[54–58], but also to operate by a docking mechanism[6]. Here, ~33% of the phospho-peptides within the DDK cluster harbored D, E, S or T at the +1 position (Fig. 1e), while ~37% of the phospho-peptides contained a proline (P) at the +1 position (Fig. 1e). Since S/T-D/E/S/T was found in 32% of all phospho-peptides and S/T-P in 33%, consensus sequence does not appear to be the only driver of DDK phosphorylation in vivo. These data are consistent with a proposed substrate-docking mechanism[6] and a recent phospho-proteomic study of DDK substrates in human cells[59]. To analyze whether DDK may operate on a pathway level, we performed gene ontology (GO) analysis of biological processes (BP) and identified several GO terms to be enriched (Supplementary Data 1) including DNA repair (GO:0006281, Fig. 1f), double-strand break repair (GO:0006302, Supplementary Fig. 1e), DNA damage response (GO:0006974, Supplementary Fig. 1f), and chromatin remodeling (GO:0006338, Supplementary Fig. 1g). Within the DDK-dependent phosphorylation substrates, we found known DDK substrates, such as Mms4, Mus81 and Rif1[15,60,61], as well as Rad18, for which the human ortholog have been shown to be phosphorylated by DDK[62]. Furthermore, we also identified DDK-dependent phosphorylation of Mre11 and Xrs2 (Fig. 1f and Supplementary Fig. 1h) at sites that previously were suggested to be phosphorylated by CDK[63]. While further clarification is needed whether those sites are phosphorylated by CDK or DDK or both, we did not follow-up on those results, since mutational analysis had suggested that phosphorylation of these sites was not involved in HR[63]. Next to Mre11 and Xrs2, the phospho-proteomic analysis indicated DDK-

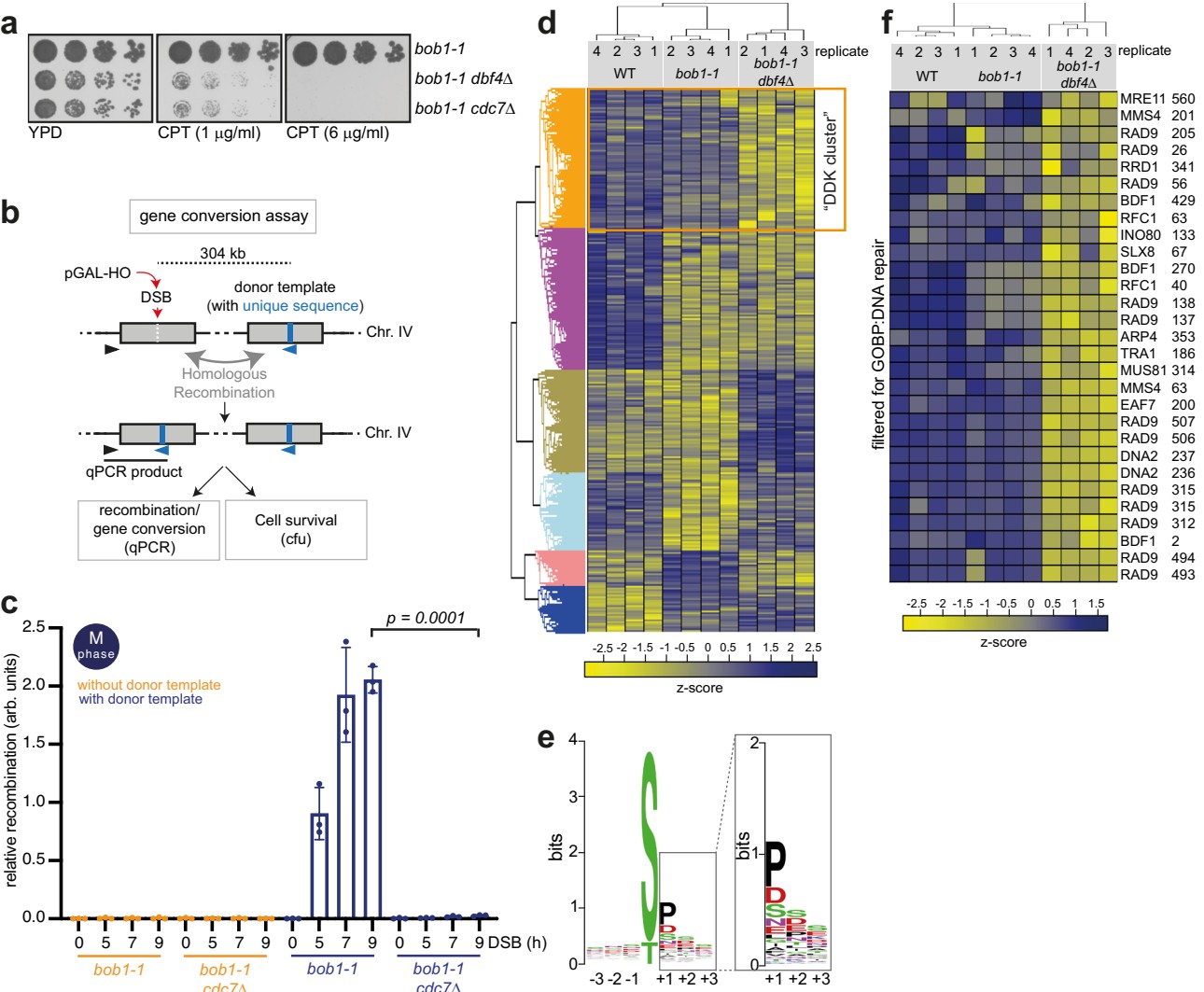

**Fig. 1 | DDK phosphorylates HR proteins and is required for HR. a** DDK mutants are hypersensitive to camptothecin (CPT). Five-fold serial dilutions of *bob1-1*, *bob1-1 dbf4Δ*, and *bob1-1 cdc7Δ* strains were spotted on YPD or YPD containing CPT at the indicated concentrations. Data are representative of *n* = 2 biological replicates. **b** Scheme of the gene conversion assay used to study HR rates. A DSB generated by the HO endonuclease at 491 kb (Chr. IV) can be repaired using a homologous donor template at 795 kb (Chr. IV). The latter carries a unique 23-bp sequence, allowing for quantification of the recombination products via qPCR; arrowheads denote PCR primer locations. Gene conversion using the donor template disrupts the HO cut site, allowing cell survival to the chronic induction of the endonuclease as an alternative readout. 'cfu' stands for colony-forming unit. **c** DDK is required for HR. qPCR analysis of HR using system as in (**b**) in cells arrested in M-phase. Cells lacking the donor template are used as control. *n* = 3, box plot shows mean with values of biological replicates, error bars denote SD. Reported *p*-values were calculated using a two-tailed unpaired *t*-test. See also Supplementary Fig. 1a, b. **d** Class-I peptides derived from phospho-proteomic experiment were subjected to analysis of variance (ANOVA) test, with permutation-based false discovery rate (FDR) cutoff of 0.05. ANOVA significant phospho-peptides were then subjected to a hierarchical clustering in Perseus (v1.6.5.0). The calculated z-scores are shown in the heat-map and the different clusters are highlighted. The DDK cluster shows specific phospho-peptides downregulated in *bob1-1 dbf4Δ* cells. *n* = 4 biological replicates. See also Supplementary Fig. 1c–h. **e** Motif sequence generated for phospho-peptides enriched in the DDK cluster showing the 3 positions upstream and downstream the modified S/T. **f** Heat-map depicting the z-score, highlighting phospho-peptides from the DDK cluster after filtering in Perseus (v1.6.5.0) for GOBP:DNA repair (GO: 0006281). Gene name and modified residues are reported. GOBP = Gene Ontology Biological Process. Source data are provided as a Source Data file.

dependent phosphorylation occurring on other resection proteins, namely Dna2, Fun30, Rad9 (Fig. 1d, f, Supplementary Fig. 1e–g). Interestingly, these three proteins have previously been shown to be regulated by CDK phosphorylation[48,50,51,64,65].

## DDK promotes DNA end resection and phosphorylates resection nucleases

We analyzed the involvement of DDK in DNA end resection in yeast using three approaches. First, we used a genetic assay system for single-strand annealing (SSA) repair[66]. Specifically, the SSA tester strain (deleted for the *RAD51* recombinase) can survive chronic induction of an HO-induced DSB in the *LEU2* open reading frame, by resecting 25 kb

of DNA to reveal a region homologous to the cut site (Fig. 2a). Deletion of *CDC7* in the *bob1-1* background renders cells deficient for SSA (Fig. 2b), consistent with resection requiring DDK. Second, we measured resection of an HO-induced, non-repairable DSB using a physical assay. Specifically, we monitored the appearance of single-stranded DNA (ssDNA) in M-phase-arrested cells using chromatin immunoprecipitation (ChIP) of the ssDNA-binding protein RPA[67,68]. When we measured the accumulation of RPA-bound ssDNA next to an HO-induced DSB at the *MAT* locus by qPCR, we found that 4 h after DSB induction, *bob1-1 cdc7Δ* cells had resected less DNA compared to the *bob1-1* control (Supplementary Fig. 2a). As an alternative system to interfere with DDK function, we generated an auxin-inducible degron

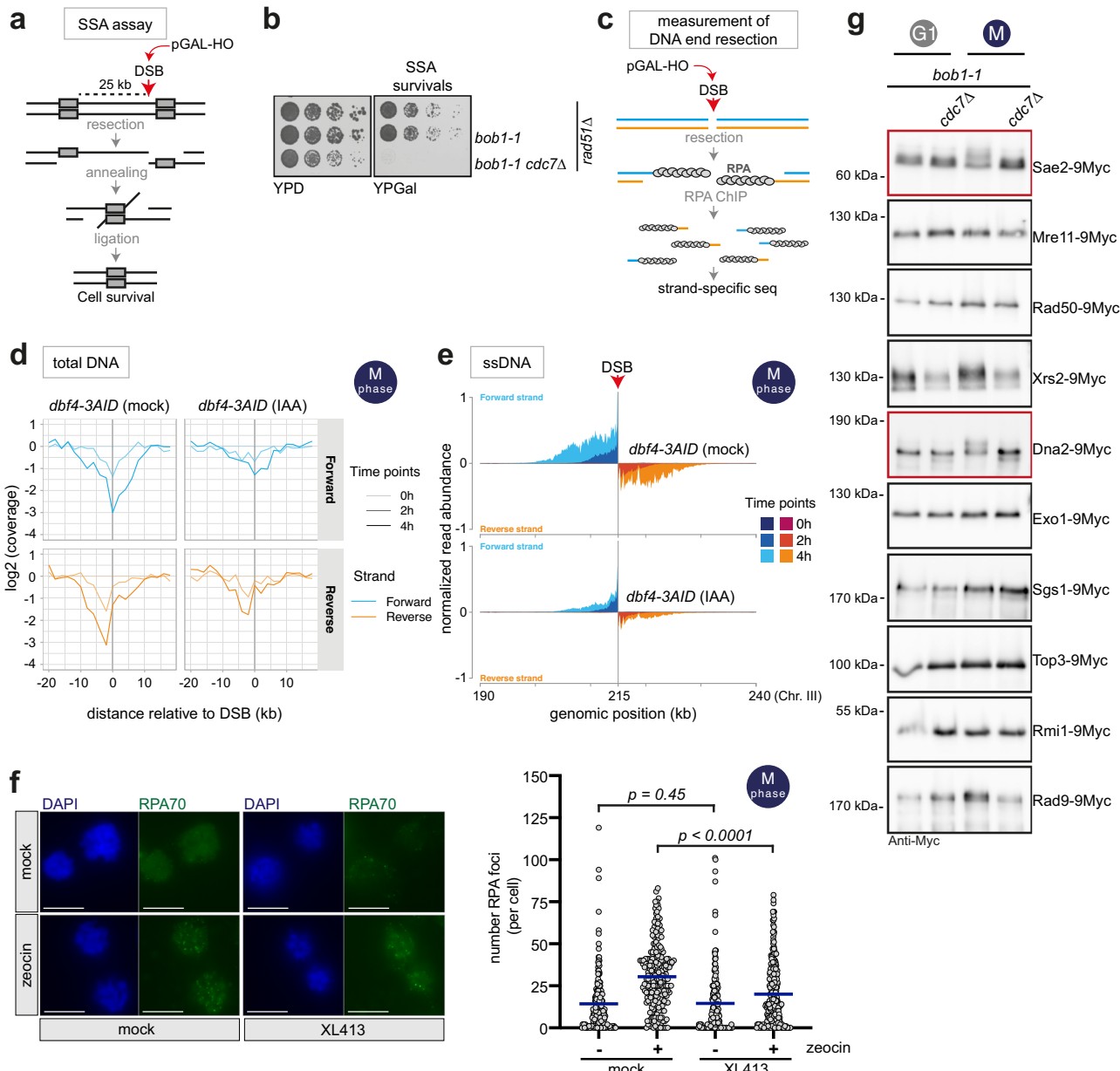

**Fig. 2 | DDK promotes DNA end resection and phosphorylates resection nucleases. a** A DSB generated by the HO endonuclease in the *LEU2* open reading frame can be repaired via single-strand annealing (SSA) upon extensive resection that reveals a region of homology 25 kb distant from the DSB, followed by annealing of the homologous sequences and ligation. *RAD51* is deleted to suppress HR. **b** Five-fold serial dilutions of reported strains were spotted on YPD (as control) or YPGal (for chronic pGAL-HO-induction) plates. Data are representative of *n* = 3 biological replicates. **c** Strand-specific loss of DNA and enrichment of RPA-bound ssDNA is used to measure resection at an unrepairable DSB, generated by the HO endonuclease at the *MAT* locus. **d**, **e** DDK is required for efficient DNA end resection. **d** Total DNA shows preferential loss of 5′ DNA strands 2 and 4 h after DSB induction. Loss of DNA is less pronounced after IAA-induced degradation of Dbf4-3AID. Strand-specific read coverage is normalized to read coverage before DSB induction. **e** Strand-specific accumulation of 3′-ssDNA enriched by RPA-ChIP is diminished after IAA-induced degradation of Dbf4-3AID compared to mock-treated cells. RPA-ChIP signals at the DSB are normalized to DSB-independent RPA signals

occurring throughout the genome. Data are representative of *n* = 4 biological replicates. See also Supplementary Fig. 2e, f. **f** DDK inhibition leads to defective resection in human cells. M-phase-arrested U2OS cells were treated with the DDK inhibitor XL413 (or mock-treated) before DNA damage induction with zeocin (or mock-treatment). RPA foci (RPA70 subunit) were counted as proxy for resection. Left, representative images of cells (DAPI, RPA70). Scale bars: 10 μm. Right, quantification of RPA foci number per cell. Blue bars represent the mean. Reported *p*-values were calculated using a two-tailed Mann-Whitney test. Mock/mock *n* = 171 cells, mock/zeocin *n* = 230 cells, XL413/mock *n* = 169 cells, XL413/zeocin *n* = 265 cells; pooled from *n* = 2 biological replicates. See also Supplementary Fig. 2g–j. **g** Sae2 and Dna2 display a cell cycle and DDK-dependent shift in electrophoretic mobility. Cells expressing 9Myc-tagged resection proteins as indicated were arrested either in G1 or M-phase and samples run on gels. Data are representative of *n* = 2–4 biological replicates. See also Supplementary Fig. 2k. Source data are provided as a Source Data file.

(AID) strain[69,70], which allowed conditional depletion of Dbf4-3AID (specifically, three copies of miniAID tag) by addition of Indole-3-acetic acid (IAA) or 1-Naphthaleneacetic acid (NAA) to the growth medium (Supplementary Fig. 2b, c). IAA was added to nocodazole-arrested cells

to deplete Dbf4-3AID prior to DSB induction and such DDK-depleted cells showed reduced amounts of resection of an HO-induced DSB, when compared to mock-treated cells (Supplementary Fig. 2d). Third, we followed resection using strand-specific sequencing[68], which

allowed us to measure resection by two distinct read-outs – DNA loss and accumulation of RPA-covered ssDNA (Fig. 2c–e, Supplementary Fig. 2e, f). Specifically, we sequenced genomic DNA after preparation of strand-specific NGS libraries using ssDNA-ligation from cells in which a site-specific, non-repairable DSB, was induced at the *MAT* locus using HO endonuclease. Additionally, we analyzed purified RPA-bound ssDNA using RPA-ChIP as above. For genomic DNA, we could clearly detect strand-selective resection of 5' DNA ends by loss of DNA signal within 15 kb on both sides of the DSB, which progressed into undamaged chromatin over time as expected for resection (Fig. 2d). When we acutely depleted cells of Dbf4-3AID prior to DSB induction, there was reduced DNA loss indicating a resection defect (Fig. 2d). An involvement of DDK was further supported by our measurement of ssDNA using RPA-ChIP followed by strand-specific sequencing (Fig. 2e). Here, we used normalization to HO-independent ssDNA RPA peaks in order to quantitatively compare between samples. We observed an RPA-ssDNA signal on forward and reverse strands on both sides of the DSB that is consistent with the preferred 5' strand degradation by the resection nucleases. Notably, when we compared *dbf4-3AID* cells treated with IAA with mock-treated cells, we found a reduction of the RPA-ssDNA signal in the absence of DDK (Fig. 2e). Altogether these data indicate that DDK promotes efficient DNA end resection.

Next, we aimed to test evolutionary conservation and whether DDK also regulates resection in human cells. Therefore, we used the U2OS osteosarcoma cell line and measured resection of DSBs induced by zeocin. In order to rule out confounding effects of DDK's S-phase function, we induced M-phase arrest by nocodazole (Supplementary Fig. 2g). In human cells HR is down-regulated in M-phase, but DNA end resection and RPA recruitment are active[71]. We used chemical inhibition of DDK by XL413 (Supplementary Fig. 2h, i[72]), and treated cells with 200 μg/ml zeocin for 2 h. In these experiments, we measured the appearance of RPA foci in the nucleus as proxy of resection. Quantification of RPA foci showed down-regulation of resection after XL413 treatment consistent with previous observations[73] (Fig. 2f). We also checked induction of DNA damage using γH2AX level, but these were not influenced by DDK inhibition (Supplementary Fig. 2h, j). These data indicate that DDK promotes resection also in human cells and hence this DDK function is likely conserved throughout eukaryotic evolution.

To identify DDK phosphorylation targets in the DNA end resection pathway we decided to complement our phospho-proteomic approach and specifically test for cell cycle-dependent phosphorylation of the core resection machinery by Western Blot. We decided on this dual strategy since our phospho-proteomics approach measured >11.000 of the estimated 45.000 budding yeast phosphorylation sites, which is a known limitation of the method[74] (Fig. 1d). We directly tested Sae2, Mre11, Rad50, Xrs2, Sgs1, Top3, Rmi1, Dna2, Exo1 and Rad9 by adding a 9Myc-tag and testing for phosphorylation-dependent upshift in SDS-PAGE in *bob1-1* and *bob1-1 cdc7*Δ backgrounds. When comparing cells arrested in G1-phase to cells arrested in M-phase, either with or without DDK, we observed a cell cycle-dependent shift in mobility for both Sae2 and Dna2 (Fig. 2g, Supplementary Fig. 2k). Notably, this shift was dependent on DDK (Fig. 2g, Supplementary Fig. 2k) and we confirmed that it was caused by protein phosphorylation, as it was sensitive to treatment with λ-phosphatase (Supplementary Fig. 2l). We also tested CDK-dependence and performed similar experiments in *cdc28-as1* cells (in which CDK can be inhibited using 1-NM PP1[75]). In case of Sae2 we still observed an upshift upon CDK inhibition, albeit reduced, while for Dna2 the shift was lost (Supplementary Fig. 2m–p). Therefore, DDK targets at least two resection nuclease complexes, the Sae2-MRX and STR-Dna2, offering a plausible model for its role as an activator of resection.

## DDK regulates short-range resection via phosphorylation of Sae2

Sae2 together with the MRX complex is important for the initiation of DNA end resection[76,77]. Sae2 was previously shown to be targeted by CDK, and in vivo and in vitro analysis showed CDK-mediated phosphorylation of Sae2 to be required - but not sufficient - for DNA end resection[25,45,46]. To test if DDK-mediated phosphorylation of Sae2 could directly stimulate the activity of the Sae2-MRX complex, we phosphorylated recombinant, λ-phosphatase-treated, Sae2 in vitro using purified DDK (Fig. 3a), reconstituted the Sae2-MRX complex and tested for endonuclease activity. We used a linear DNA substrate, where the DNA ends were protected from exonucleolytic attack through streptavidin moieties, allowing the measurement of the endonuclease activity[25,46] (Supplementary Fig. 3a). Interestingly, we observed that DDK-phosphorylated Sae2 was able to stimulate the endonuclease activity of Sae2-MRX approximately 2-3-fold compared to non-phosphorylated Sae2 (Fig. 3b). We also recapitulated the already known CDK-mediated phosphorylation of Sae2 (Fig. 3a) and Sae2-MRX activity was slightly higher after CDK phosphorylation of Sae2 compared to DDK phosphorylation (Supplementary Fig. 3b). The CDK-mediated regulation of Sae2 relies on phosphorylation of serine 267[25,45,46]. We therefore combined a CDK phospho-mimetic version of Sae2 (Sae2-S267E) with DDK phosphorylation. While Sae2-S267E partially stimulated the endonucleolytic activity of the MRX complex[45,46], we observed an additional enhancement of Sae2-MRX endonuclease activity when Sae2-S267E was phosphorylated by DDK (Fig. 3c). Collectively, these data suggest that DDK regulates Sae2-MRX and stimulates activity of the Mre11 complex. Furthermore, CDK and DDK phosphorylation appear to act additively to stimulate Sae2-MRX activity.

The Sae2-induced endonucleolytic activity of the MRX complex is essential to cleave DNA-hairpin structures, and thus remove secondary DNA structures from DSBs[78]. To translate our in vitro findings to an in vivo set-up we therefore used a genetic assay to monitor hairpin cleavage-mediated induction of recombination in a DDK mutant[78] (Fig. 3d). In *bob1-1 cdc7*Δ cells, we observed a ~5-fold reduced recombination rate, compared to a ~30-fold reduction observed in the Mre11 nuclease deficient *mre11-H125N* mutant (Fig. 3e).

Additionally, we monitored resection of a non-repairable HO-induced DSB in cells defective for long-range resection, where Sae2-MRX was the only resection nuclease (*exo1*Δ *sgs1-AID dna2-AID*[79,80]; Fig. 3f). In this scenario, we measured short-range resection at +98 bp and −120 bp distance from the DSB[81]. Depleting DDK in IAA-treated *dbf4-3AID* (*exo1*Δ *sgs1-AID dna2-AID*) cells led to a defect in Sae2-MRX mediated short-range resection (Fig. 3g, h, Supplementary Fig. 3c, d). We also tested the CDK phosphorylation-deficient *sae2-S267A* mutant and observed that resection by Sae2-MRX was largely deficient in *sae2-S267A* and *sae2-S267A dbf4-3AID* strains (Supplementary Fig. 3e–g).

To clarify the functional relationship between CDK and DDK phosphorylation of Sae2, we sought to identify sites on Sae2 that could be targeted by DDK. In the absence of phospho-proteomics data, we followed the idea that DDK preferentially phosphorylates S (or T) followed by D, E, phospho-S or T in position +1[54–58]. We generated a Sae2-6A mutant by mutating all S/T-D/E sites to alanine, and a Sae2-14A mutant by additionally targeting all S/T-S/T sites (Fig. 3i, Supplementary Fig. 3h). We then complemented *sae2*Δ cells with plasmids carrying either Sae2, Sae2-6A or Sae2-14A versions, all harboring a 9Myc-tag (Supplementary Fig. 3i). Notably, in the Sae2-6A mutant the cell cycle and DDK-dependent phospho-shift was lost, suggesting that DDK is targeting S/T-D/E sites in Sae2 (Fig. 3j, Supplementary Fig. 3j, k). Consistently, we observed a pronounced defect in the phosphorylation of the Sae2-14A mutant protein (Supplementary Fig. 3j, k). We caution, however, that this mutant showed reduced Sae2 protein level, which may suggest a defect in expression, folding, localization or stability of the Sae2-14A protein (Supplementary Fig. 3i, k) and

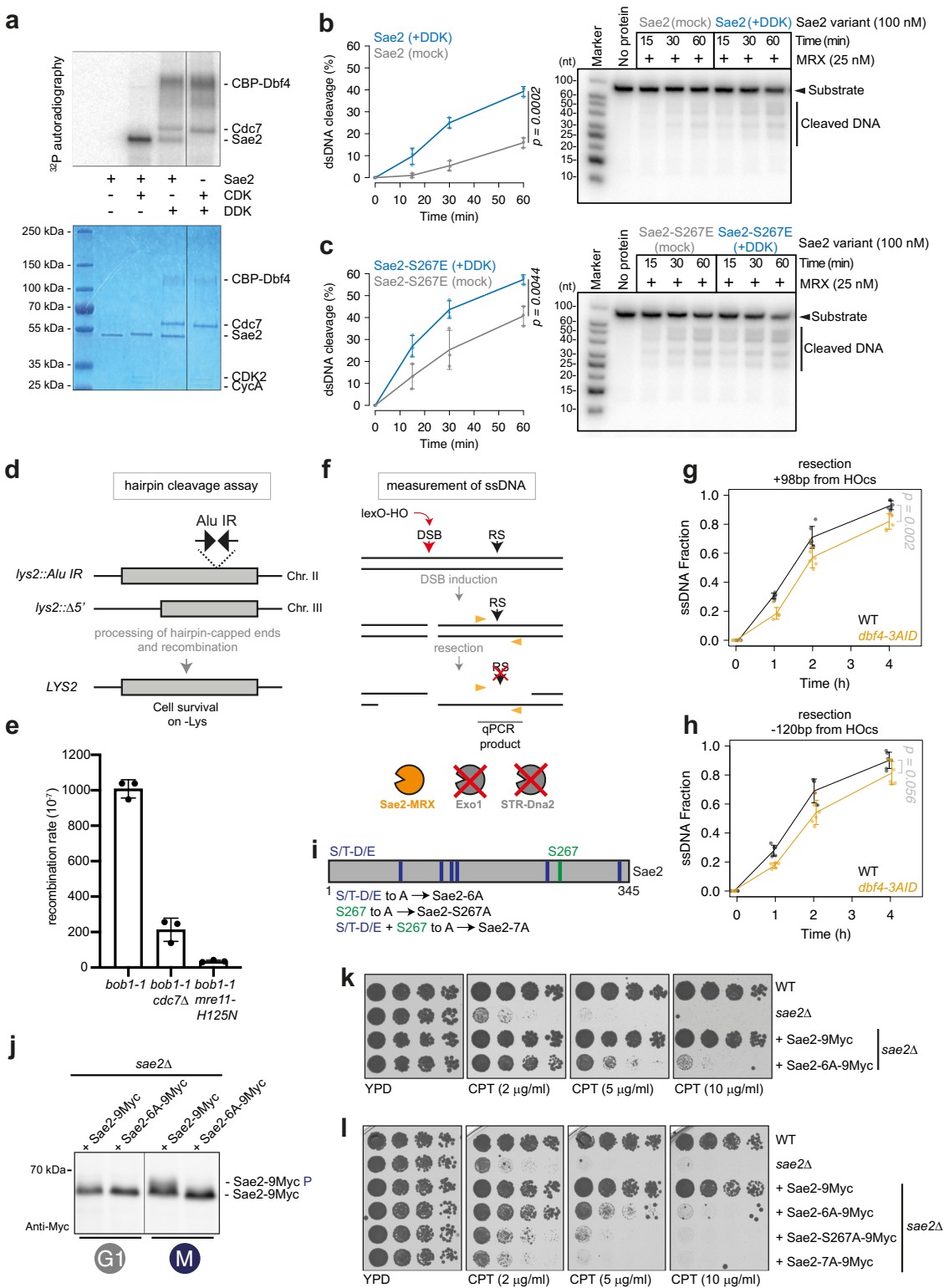

therefore did not analyze this mutant further. To correlate the putative DDK target sites to Sae2's functions in DNA repair, we tested hypersensitivity to CPT of *sae2* mutant strains. We observed a pronounced hypersensitivity of *sae2-6A* cells (Fig. 3k). To analyze the functional relationship between CDK and DDK phosphorylation, we combined the *sae2-6A* mutation with the *sae2-S267A* mutation, generating a *sae2-7A* mutant strain (Fig. 3i, Supplementary Fig. 3l). Already the *sae2-*

*S267A* single mutant strain showed pronounced hypersensitivity towards CPT that was similar to the *sae2Δ* deletion mutant (Fig. 3l[45]). The phenotype of the *sae2-S267A* mutant was stronger compared to the *sae2-6A* mutant and the combined *sae2-7A* mutant showed a small increase in CPT sensitivity compared to the *sae2-S267A* strain (Fig. 3l). This suggests that CDK and DDK phosphorylation play at least partly independent roles in the control of the Sae2-MRX complex.

**Fig. 3 | DDK regulates short-range resection via phosphorylation of Sae2. a** A model CDK (mammalian CDK2-CycA) and budding yeast DDK can phosphorylate Sae2 in vitro. Top, autoradiography monitoring incorporation of radioactive phosphate; Bottom, same gel stained with Coomassie Blue. Note that DDK can auto-phosphorylate, as previously shown[15,85]. Data are representative of $n = 2$ independent experiments. **b, c** Sae2 (**b**) or Sae2-S267E (**c**) were phosphorylated by DDK or mock-treated, and added to the MRX complex to monitor endonucleolytic clipping of DNA. Left, quantification of cleavage products resolved on gels such as shown on the right. $n = 3$ independent experiments, shown is the mean with values of replicates, error bars denote SD. Reported $p$-values were calculated using a two-tailed unpaired $t$-test. **d** Inverted Alu repeats (Alu-IR) at the *LYS2* locus induce Sae2-MRX endonucleolytic cleavage and recombination with a locus carrying a truncated version of *LYS2* (*lys2::Δ5'*). **e** Recombination rates were calculated using a fluctuation analysis. $n = 3$, 7–8 fluctuations used per replicate per strain. Box plot shows mean with values of biological replicates, error bars denote SD. **f** ssDNA accumulation is monitored by resistance to restriction enzyme cleavage after DSB

induction via the HO nuclease at the *MAT* locus. The *exo1Δ sgs1-AID dna2-AID* background make the assay specific for Sae2-MRX-dependent short-range resection. RS = restriction site. **g, h** Depletion of Dbf4 induces defect in Sae2-MRX mediated resection. ssDNA accumulation upon DSB induction measured 98 bp downstream (**g**) and 120 bp upstream (**h**) the DSB via qPCR after digestion with restriction nucleases RsaI and MseI, respectively. $n = 6$ biological replicates, shown is mean with values of replicates, error bars denote SD. Reported $p$-values were calculated using a two-tailed unpaired $t$-test. See also Supplementary Fig. 3c, d. **i** Scheme of Sae2 highlighting S/T-D/E sites and S267. **j** Cells of indicated strains were arrested in G1 or M-phase to monitor the Sae2 phospho-shift. Data are representative of $n = 3$ biological replicates. See also Supplementary Fig. 3i. **k, l** Five-fold serial dilutions of indicated strains were grown on YPD plates or YPD plates supplemented with CPT at the indicated concentrations. Data are representative of $n = 3$ biological replicates. See also Supplementary Fig. 3i, l. Source data are provided as a Source Data file.

Our in vitro and in vivo analyses together suggest that DDK phosphorylates Sae2 and mutation of putative DDK target sites causes a DSB repair defect. Moreover, our data indicate that DDK phosphorylation is required for efficient short-range resection by the Sae2-MRX complex and DDK activates Sae2-MRX in a manner that is non-redundant with CDK phosphorylation.

## DDK regulates long-range resection via phosphorylation of Dna2

Similarly to Sae2, the long-range resection nuclease Dna2 appears to be phosphorylated by both CDK and DDK (Figs. 1f and 2g[48]). In contrast to Sae2, however, CDK and DDK phosphorylation appear to be linked. First, CDK and DDK were both required for the cell cycle-specific phosphorylation shift of Dna2 (Fig. 2g, Supplementary Fig. 2p). Second, we mapped a critical DDK phosphorylation site (S236) to the immediate proximity of a previously identified CDK phosphorylation site (S237[48,82]). Our phospho-proteomic analysis suggested that S236 phosphorylation is dependent on DDK (Fig. 4a). Moreover, also S237 phosphorylation may be impaired in *bob1-1 dbf4Δ* cells (Supplementary Fig. 4a). Therefore we directly checked phospho-peptide evidences, and identified double-phosphorylated peptides (S236-phospho/S237-phospho) in all replicates of WT and *bob1-1* controls (Supplementary Fig. 4b). In contrast, we identified only singly phosphorylated peptides in all replicates of the *bob1-1 dbf4Δ* strain (Supplementary Fig. 4b, for each peptide phosphorylation assigned with 80 to 95% probability to serine 237). We conclude that DDK is required for phosphorylation specifically of S236. We next mutated S236 to a non-phosphorylatable alanine and generated a *dna2-S236A* mutant, and tagged it with 9Myc (Supplementary Fig. 4c). Interestingly, we observed loss of the cell cycle-dependent phospho-shift of Dna2 in *dna2-S236A* cells, suggesting that S236 is a key phosphorylation site in Dna2 (Fig. 4b, c, Supplementary Fig. 4d, e). Three CDK target sites (T4, S17, S237 (Fig. 4b)) were previously mapped on Dna2 and a *dna2-3A* mutant strain was hypersensitive to CPT, in a background in which the second long-range resection pathway was abolished by deleting *EXO1*[48]. The *dna2-S236A* single mutant did not show hypersensitivity to CPT (Fig. 4d). However, we observed that the *dna2-S236A exo1Δ* double-mutant strain showed an increased sensitivity to CPT compared with both single mutant strains (Fig. 4d). We therefore conclude that DDK-mediated phosphorylation of Dna2 is critical for efficient DSB repair.

To check more directly if DDK is important for Dna2-mediated long-range resection, we investigated resection of a non-repairable HO-induced DSB under conditions where resection entirely relies on STR-Dna2 (*exo1Δ mre11-H125N*, Fig. 4e, Supplementary Fig. 4f, g). Note that Sae2-MRX is partially dispensable for resection of clean nuclease-induced DSBs[83]. In this context, we measured accumulation of ssDNA at 640 bp, 2.5 kb, and 5 kb distance from the DSB[79]. When we abolished

DDK function using the *dbf4-3AID* degron, we found a mild, but reproducible defect in Dna2-dependent long-range resection (Fig. 4e).

We reasoned that in the absence of DDK mild defects in Sae2-MRX-dependent short-range resection (Fig. 3g, h) and STR-Dna2-dependent long-range resection (Fig. 4e) could add up to generate the rather pronounced defects in overall resection (Fig. 2a–e). Therefore, we investigated resection in a context where it relied on Sae2-MRX for short-range resection and STR-Dna2 for long-range resection (*exo1Δ*, Fig. 4f, Supplementary Fig. 4f, h). When we depleted DDK by the *dbf4-3AID* degron in the *exo1Δ* background, we found a pronounced resection defect (Fig. 4f). We therefore conclude that DDK targets resection in a two-pronged manner by stimulating Sae2-MRX-dependent short-range resection and STR-Dna2-dependent long-range resection. We reason that due to partial compensation of resection by the different nucleases, partial defects in phosphorylation-dependent activation of the Mre11 complex or Dna2 will synergize upon combination, yielding the overall pronounced resection defect observed in the absence of DDK.

## Synthetic activation of DDK activates limited DNA end resection and homologous recombination in G1 cells

Our data identifies DDK as a new cell cycle regulator of DNA end resection. Since both resection and DDK are largely inactive in G1 cells, we wondered whether we could engineer DDK to synthetically activate resection and HR in G1 cells. Specifically, DDK is cell cycle controlled by cell cycle-regulated expression and degradation of Dbf4, which is largely absent from G1 cells[84–87]. To uncouple DDK from its endogenous cell cycle control, we overexpressed a codon-optimized version of *DBF4* (stabilized by D-box mutation) as well as *CDC7* from bidirectional *pGAL1-10* promoter[88] (Fig. 5a). We arrested cells in the G1-phase of the cell cycle using α-factor and induced DDK expression, which triggered Sae2 phosphorylation as seen by phosphorylation-dependent shift in electrophoretic mobility (Fig. 5b, Supplementary Fig. 5a). These data indicated that synthetic activation of DDK leads to premature phosphorylation of Sae2.

We next tested if synthetic activation of DDK in G1 could lead to activation of DNA end resection. We induced a non-repairable DSB at *MAT* using the pGAL-HO system and tested for resection using strand-specific RPA-ChIP-seq in G1-arrested cells (Fig. 5c, Supplementary Fig. 5b, c). In the WT G1 cells, as expected, resection was very limited. However, when we synthetically activated DDK (*GAL-DDK*), resection was notably enhanced. We also tested the CDK phospho-mimetic mutant of *SAE2* (*sae2-S267E*). Here, G1-arrested *sae2-S267E* cells increased resection compared to WT cells, as well, and we could observe an even higher RPA enrichment when *GAL-DDK* expression was combined with the *sae2-S267E* phospho-mimetic allele (Fig. 5c). We note, however, that compared to a DSB in M-phase cells, resection around the DSB was less pronounced in all G1 samples (Supplementary

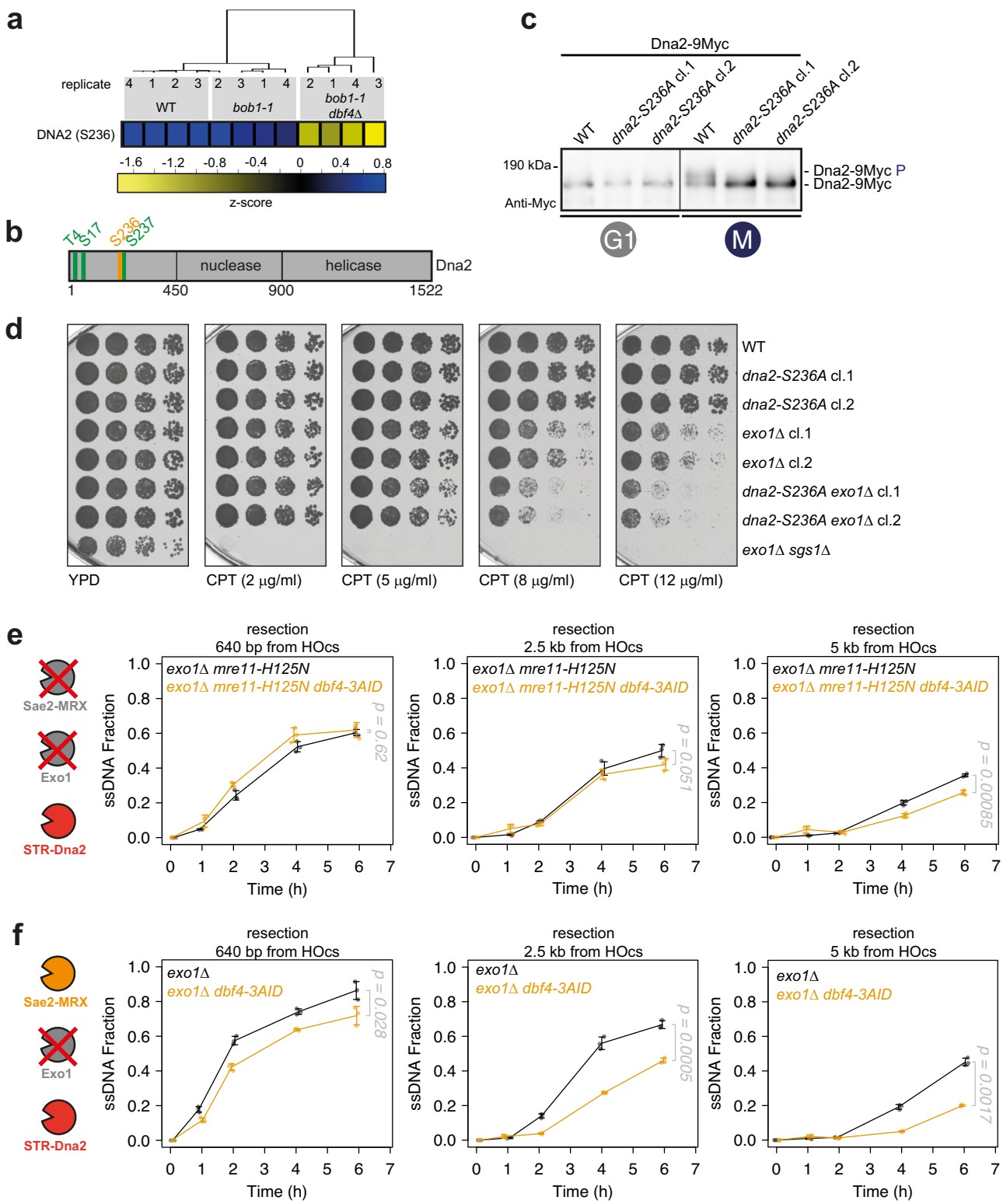

Fig. 5d). Overall, we conclude that activation of DDK in G1 partially bypasses the inhibition of resection, and that CDK and DDK phosphorylation appear to act independently of each other.

Given that synthetic activation of DDK in G1 allows a limited resection, we wondered if the activation of DDK in G1 would allow recombination-mediated repair. We employed the gene conversion assay (Fig. 1b) in G1-arrested cells that were kept in arrest throughout the experiment (Fig. 5d, Supplementary Fig. 5e, f). In this assay, synthetic activation of DDK in G1 led to 4-fold increased HR (Fig. 5d)

compared to G1-arrested WT cells. In contrast, *sae2-S267E* cells did not show a pronounced enhancement of HR (Fig. 5d). We also observed that DDK-dependent stimulation of HR was 10-fold lower compared to what we observed in M-phase-arrested cells (compare Fig. 5d and Fig. 1c). In the literature, the strongest effects on resection and HR were previously observed in NHEJ-deficient mutants of the Ku-complex[42,89]. We therefore compared the effect of DDK-mediated activation of HR to that of the *yku80Δ* mutant (Fig. 5e, Supplementary Fig. 5g, h). Interestingly, we could not detect a significant difference

**Fig. 4 | DDK regulates long-range resection via phosphorylation of Dna2.**
**a** Serine 236 of Dna2 is phosphorylated in a DDK-dependent manner. Data from phospho-proteomic experiment of Fig. 1d. Heat-map depicts the z-score. See also Supplementary Fig. 4a, b. **b** Scheme of Dna2 phosphorylation sites. Green: CDK target sites previously identified[48]; Orange: DDK target site S236, within S/T-S/T motif. **c** Serine 236 phosphorylation contributes to DDK-dependent Dna2 phosphorylation shift in vivo. Cells of the indicated strains were arrested in G1 or M-phase and samples collected and loaded on a gel to monitor the Dna2 phosphoshift. Data are representative of *n* = 3 biological replicates. See also Supplementary Fig. 4c, d. **d** *dna2-S236A* strain is sensitive to CPT, when *EXO1* is deleted. Five-fold serial dilutions of WT, *exo1Δ*, *dna2-S236A*, and *dna2-S236A exo1Δ* strains were grown on YPD plates or YPD plates supplemented with CPT at the indicated

concentrations. Data are representative of *n* = 3 biological replicates. **e** Depletion of Dbf4 induces defects in STR-Dna2-mediated long-range resection. ssDNA accumulation upon DSB induction was measured via qPCR after digestion with the RsaI restriction nuclease at sites with indicated distances from the DSB. *n* = 3 biological replicates, shown is mean with values of replicates, error bars denote SD. Reported *p*-values were calculated using a two-tailed unpaired *t*-test. See also Supplementary Fig. 4f, g. **f** DDK-dependent resection defect accumulates in the absence of *EXO1*. Resection was measured as in (**e**), but in a background where resection was carried out by Sae2-MRX and STR-Dna2. *n* = 3 biological replicates, shown is mean with values of replicates, error bars denote SD. Reported *p*-values were calculated using a two-tailed unpaired *t*-test. See also Supplementary Fig. 4f, h. Source data are provided as a Source Data file.

between HR in *yku80Δ*, *GAL-DDK* or the corresponding double mutant (Fig. 5e), suggesting that all conditions activate HR via the same mechanism, likely initiation of DNA end resection. We confirmed that these effects were not caused by leakage from G1-arrest using FACS-based DNA content measurements and budding index (Supplementary Fig. 5e, g, i) and ruled out that they would arise from differences in the number of G1 cells in the investigated cell population (Supplementary Fig. 5f, 5h). Moreover, stimulation of HR was specific for G1 cells and not seen in cells arrested in M-phase (Supplementary Fig 5j, k). This study therefore not only identifies DDK as cell cycle regulator of DNA end resection and HR, but we could also employ this knowledge to engineer genetic conditions that influence the endogenous DSB repair pathway choice.

## Discussion

Cell cycle state is known to be a major determinant of DSB repair[44]. Here, we uncovered a new layer of this cell cycle regulation that involves the cell cycle kinase DDK (model in Fig. 5f). We find that DDK is required for efficient DNA end resection and HR in budding yeast and that its function as regulator of resection is conserved to humans. Previous work uncovered that resection is cycle-controlled by CDK (Cdc28 in yeast, CDK1/2-CyclinA/B in humans)[43,44]. In yeast, CDK phosphorylation targets a set of resection factors and regulators including Sae2, Dna2, Rad9, and Fun30[45,48,50,51,64,65]. Interestingly, phospho-proteomic analysis of DDK substrates and directed screening of phosphorylation of resection enzymes revealed that DDK acts on the same set of proteins, suggesting that these factors are key hubs for control. In this study, we focused on the phosphorylation of Sae2 and Dna2. Sae2 is the activator of the Sae2-MRX complex responsible for resection initiation (short-range resection), while STR-Dna2 is one of two long-range resection nucleases[18]. DDK phosphorylation-deficient mutants of Sae2 and Dna2 cause DNA damage hypersensitivity and, consistently, we observed that both short-range and long-range resection were impaired in the absence of DDK (Fig. 5f). Notably, defects of either Sae2-MRX or STR-Dna2-dependent resection were observed in the absence of DDK and these defects cumulated when resection was dependent on both enzymes, suggesting synergy of DDK phosphorylation on different substrates.

In this study we focused specifically on regulation of resection nucleases by DDK, which constitutes the most direct circuit of resection control. Previous work has shown, however, that DSB-surrounding chromatin plays an important role in the regulation of resection, too[38,90]. The resection-inhibitory nucleosome binder Rad9 has previously been shown to be targeted by CDK phosphorylation, as has its human ortholog 53BP1[64,65,91]. Similarly, the nucleosome remodeler Fun30 and its human ortholog SMARCAD1 are both targeted by CDK as well and this promotes resection[50,51]. Notably, our phospho-proteomics analysis identifies Fun30 and Rad9 as DDK targets. It can therefore be speculated that DDK phosphorylation of Rad9 and/or Fun30 helps to establish resection-permissive chromatin and that these factors contribute to the observed resection defects in the absence of DDK. Further studies will be needed to test this hypothesis.

We also find evidence for DDK being a regulator of DNA end resection in human cells. These data are consistent with recent studies showing that DDK is required for HR also in human cells[73,92]. DDK functions have mainly been studied in the context of DNA replication: next to the critical function for replication initiation[93], human DDK is required at replication forks throughout S-phase, where it could mediate replication fork protection and restart[59]. Interestingly, in BRCA-deficient cells, DDK appears to promote resection of DNA at reversed replication forks, possibly via regulation of MRE11[94,95]. While DDK phenotypes in the context of DNA replication are often difficult to disentangle from its replication initiation function, these data are entirely consistent with the role of DDK in promoting resection being conserved between yeast and human cells. These findings may open future avenues for therapies and, indeed, DDK inhibition enhanced anti-tumor activities of DNA-damaging based chemotherapies and was suggested as a system to chemically induce BRCAness-like phenotypes in preclinical tumor models, leading to synthetic lethality with PARP inhibitors[92]. Our data therefore provide a conceptual framework for the requirements of DDK for HR.

It is interesting to note that additional functions of DDK in response to replication stress, replication fork restart specifically, have been identified in budding yeast[96,97]. Moreover, DDK was suggested to be involved in break-induced replication (BIR)[98].

Many cell cycle-regulated processes are controlled by two or more cell cycle kinases. In particular, CDK and DDK phosphorylation were shown to cooperate on several processes, most prominently replication initiation. It is remarkable that key resection proteins appear to be phosphorylated by both kinases. However, the degree of cooperation of CDK and DDK is different for the two substrates tested, Sae2 and Dna2. DDK is known to target S/T-S/T sites that are already phosphorylated at the +1 position[55-58]. Such priming phosphorylation could be carried out by CDK and indeed CDK has been shown to act as priming kinase for DDK phosphorylation of several substrates[8-10,55,56,58]. Also for Dna2 a CDK-priming model appears to be likely for several reasons: (i) S236 is in immediate proximity to S237, which was previously identified to be phosphorylated by CDK[48,82]; (ii) CDK inhibition, DDK depletion, as well as the *dna2-S236A* all abolish the Dna2 phosphorylation shift and (iii) the phenotype of the *dna2-S236A* mutant appears to phenocopy what previously has been observed in a CDK phosphorylation-deficient *dna2-3A* mutant[48].

CDK phosphorylation of Dna2 at S17 (and T4) was shown to be required for Dna2 nuclear localization[48,99]. In contrast, CDK phosphorylation on S237 was shown to be important to regulate the recruitment of Dna2 to DSBs[48]. Given the possible interdependency of S236 and S237 phosphorylation it is therefore tempting to speculate that DDK-mediated phosphorylation of Dna2 could be involved in promoting DSB localization of Dna2 as well.

In the case of Sae2 phosphorylation, CDK and DDK appear to act independently of each other. While Sae2 carries three CDK consensus sites (S134, S179, and S267) and S134 and S179 are part of S/T-S/T-P motifs, we see that mutation of all S/T-D/E motifs (*sae2-6A*) largely abolishes the cell cycle-dependent phosphorylation shift. In the same

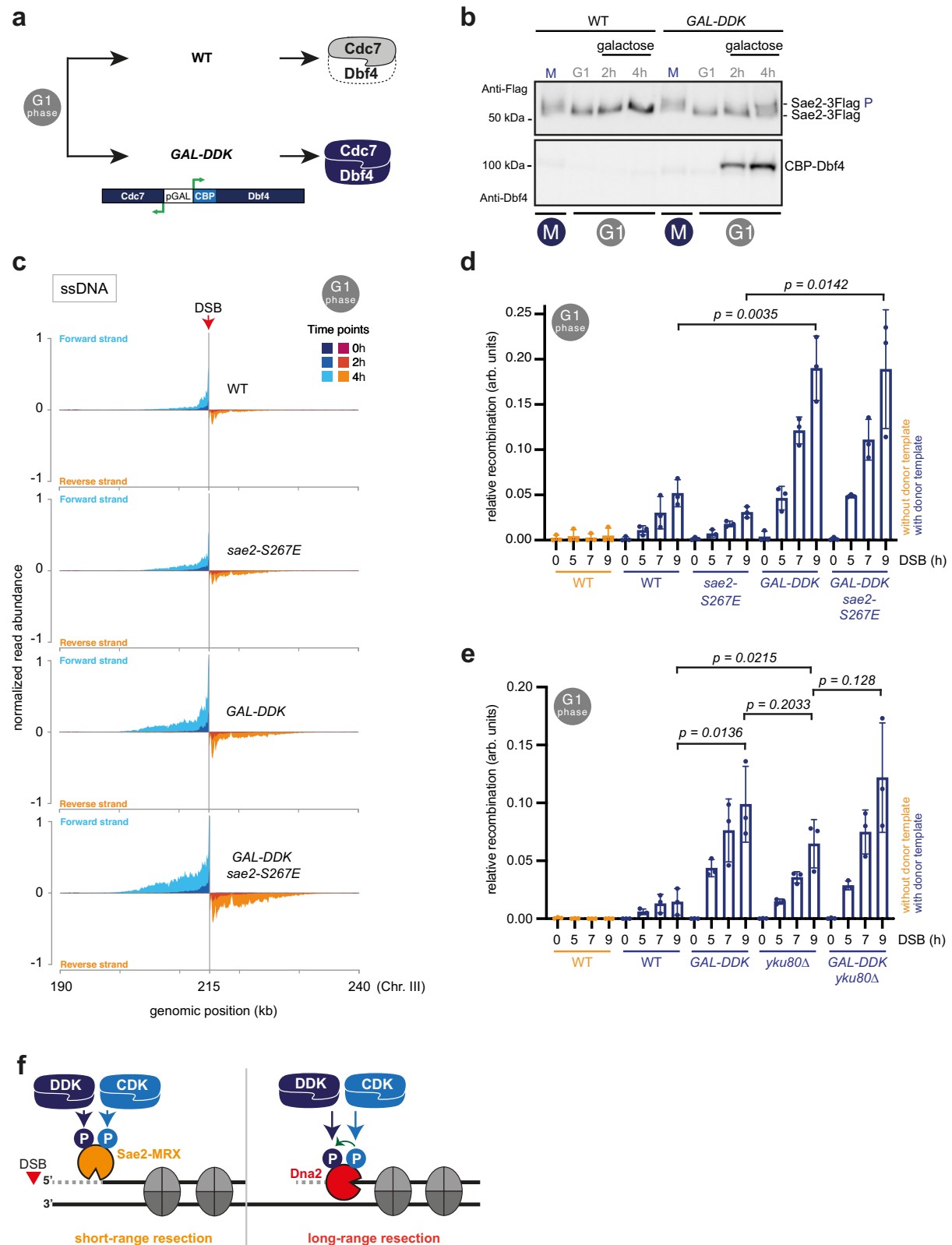

vein, depletion of DDK abolishes the Sae2 phosphorylation shift, while inhibition of CDK using *cdc28-as1* only leads to a reduction, contrary to what would be expected if CDK primed for DDK phosphorylation. Similarly, in our activity assays, both DDK and CDK phosphorylation of Sae2, as well as the Sae2-S267E phospho-mimetic protein are able to stimulate Sae2-MRX activity. For CDK phosphorylation of Sae2 (or its human ortholog CtIP) a plausible mechanism was derived from work on the ortholog Ctp1 from *S. pombe*, which defined a 15 amino acids peptide to be critical for stimulating the endonuclease activity of Mre11[100]. Notably, this peptide spans the crucial CDK sites in Sae2 and CtIP, suggesting that CDK phosphorylation acts by modulating Sae2-stimulation of Mre11 activity.

In all, the control of DSB repair pathway choice is multi-faceted and with DDK we now add an additional player controlling early steps

**Fig. 5 | Synthetic activation of DDK activates limited DNA end resection and HR in G1 cells. a** Synthetic activation of DDK in G1-arrested cells. Codon-optimized versions of DBF4 and CDC7 are expressed from bidirectional *pGAL1-10* promoter, with Dbf4 carrying D-box mutations that stabilize Dbf4[86,88]. **b** Expression of DDK induces Sae2 phosphorylation in G1. Cells of the indicated strains were arrested in G1, DDK expression was induced by addition of galactose, and samples collected and loaded on a gel to monitor the Sae2 phospho-shift. Data are representative of *n* = 2 biological replicates. See also Supplementary Fig. 5a. **c** Synthetic activation of DDK in G1 allows for limited activation of DNA end resection. Strand-specific accumulation of 3′-ssDNA enriched by RPA-ChIP in G1-arrested cells indicates resection throughout time course (2, 4 h after DSB induction) in WT, *sae2-S267E*, *GAL-DDK*, and *GAL-DDK sae2-S267E* strains. RPA-ChIP signals at the DSB are normalized to DSB-independent RPA signals occurring throughout the genome. Data

are representative of *n* = 2 biological replicates. See also Supplementary Fig. 5b-d. **d, e** Synthetic activation of DDK allows for limited recombination-mediated repair in G1. qPCR analysis of HR upon DSB induction at 491 kb (Chr. IV) using a donor template at 795 kb (Chr. IV). WT cells lacking the donor template are used as negative control. **d** Comparison of WT, *sae2-S267E*, *GAL-DDK* and *GAL-DDK sae2-S267E* strains. **e** Comparison of WT, *GAL-DDK*, *yku80Δ*, and *GAL-DDK yku80Δ* strains. *n* = 3, box plot shows mean with values of biological replicates, error bars denote SD. Reported *p*-values were calculated using a two-tailed unpaired *t*-test. See also Supplementary Fig. 5e-i. **f** Double kinase mechanism for cell cycle-regulated DNA end resection. CDK and DDK target at least two proteins, Sae2 to control Sae2-MRX-dependent short-range resection and Dna2 to control STR-Dna2-dependent long-range resection. Source data are provided as a Source Data file.

of DSB repair. A major driver for studying DSB repair pathway choice is, however, the ambition to override endogenous controls and install extrinsic controls to DSB repair. In particular, engineering HR to be the default DSB repair pathway would have tremendous impact on genome editing methods[101]. DDK inhibition was shown to negatively impact HR in human cells[73,92], but in these experiments direct effects on HR were difficult to disentangle from indirect cell cycle effects. Furthermore, timed inhibition of DDK during genome editing was also shown to be able to increase the efficiency of HR-mediated editing by inducing a slower S-phase, a cell cycle phase permissive for HR[102]. Interestingly, similar treatment performed before editing led to a reduction in the efficiency of HR-mediated editing[102].

Here, we were able to activate resection and HR in budding yeast by inducing the expression of active DDK in G1 cells, which are endogenously resection-deficient. By comparison, activation of HR upon DDK activation was as strong as in the *yku80Δ* mutant, which leads to NHEJ-deficiency, a condition known to activate HR[103]. We note, however, that even after DDK activation, HR was stimulated only to about 10% of what we observed in M-phase cells, and also resection was not fully restored. This suggests that further engineering – for example of additional CDK controls of resection as well as cell cycle-regulated expression of HR proteins – will be necessary to unleash HR for application in genome editing technologies.

## Methods

### Yeast strains
All yeast strains were constructed in the W303 background[104]. Genotypes of used strains are provided in Supplementary Data 2. Different strain numbers with the same genotype refer to two or more isolates used as biological replicates. Integrative plasmids were linearized prior to transformation and single integration of plasmids was confirmed by genotyping PCR. Gene deletions and tags were introduced using a PCR-based protocol[105,106] and confirmed by PCR (additional western blot to test expression levels). *MCM5-P83L* (*bob1-1*), *DNA2-S236 A*, and *SAE2-S267E* point mutations were introduced at the endogenous locus using a two-PCR products-based transformation[107]. *SAE2-S267A* point mutation was generated using CRISPR/Cas9-mediated genome engineering as previously described[108].

### Human cell lines
Human U2OS osteosarcoma cell line was a kind gift of Ralf Jungmann's laboratory.

### Plasmids
Genes of interest were amplified from genomic DNA of W303 or from other plasmids carrying the insert of interest, and cloned into the respective vector using the In-Fusion HD cloning kit (Clontech). *sae2-6A* and *sae2-14A* sequences were designed and generated as synthetic construct (Genscript). Sae2-6A protein contains the following S/T mutated to A: 84-130-143-149-252-335. Sae2-14A protein contains the following S/T mutated to A: 12-33-72-75-84-130-133-143-149-177-178-

252-278-335. Plasmids pJH17 and pJH19 were generated with site-directed mutagenesis of S267 to A using as template pLG36 and pLG25, respectively. After mutagenesis PCR, parental vector was degraded using DpnI (NEB). A list of plasmids used in this study is provided in Supplementary Table 1.

### Yeast cultures
Cells were grown at 30 °C in YP-medium supplemented with 2% glucose (YPD), 2% raffinose (YPR) or additional 2% galactose (YPRGal). M-phase arrest was achieved treating cultures with 5 µg/ml nocodazole (Sigma, M1404) unless stated otherwise. G1-phase arrest was achieved by treating cells with α-factor (core facility Max Planck Institute of Biochemistry or GenScript, RP01002, 0.5 µg/ml for *bar1Δ* cells or 10 µg/ml for *BAR1* cells). Note that for *BAR1* cells α-factor was freshly added every hour. Degradation of AID-tagged proteins (Dbf4, Sgs1 or Dna2) was achieved using treatment with 1 or 2 mM IAA (Sigma, I3750) or NAA (Sigma, 317918) (for 1 h), unless stated otherwise. For cells carrying the pGAL-HO system for DSB induction or the pGAL-DDK system for synthetic expression of DDK, cells were grown in YP-medium supplemented with 2% raffinose, and induction was achieved by adding 2% galactose in the culture. For cells relying on the lexO-HO system, 2 µM β-estradiol (Sigma, E8875) was added to the culture for DSB induction. For *cdc28-as1* cells, 1.5 µM 1-NM-PP1 (Cayman chemical, Cay1330) was used to inhibit CDK/Cdc28.

### Yeast growth assay
Cells were grown in YPD or YPR (supplemented with 40 mg/l adenine, for experiments relying on pGAL-mediated induction). Cultures were diluted to an OD$_{600}$ of 0.5 and 5-fold serial dilutions were prepared and spotted on plates. To test sensitivity to CPT (Sigma, C9911) cells were grown in YPD; cultures were then diluted to an OD$_{600}$ of 0.5 and 5-fold serial dilutions were spotted on YPD (as control) or YPD plus the reported CPT concentration. Plates were incubated for 2/3 days at 30 °C. For each biological replicate, at least two technical replicates were performed.

### Budding index calculation
The percentage of budded yeast cells (budding index) was calculated via microscopic inspection. Samples were collected and treated as for DNA content measurement via FACS, omitting incubation with SYTOX green (see "Flow cytometry" section) before manual bud counting.

### Flow cytometry
Around $2 \times 10^7$ yeast cells (1 OD) were harvested and the pellet resuspended in fixation buffer (70% ethanol 50 mM Tris-HCl, pH 8.0) and stored at 4 °C. Cells were then washed once with 50 mM Tris-HCl, pH 8.0 and then treated with 200 µg RNase A (Sigma, R4875; diluted in 10 mM Tris-HCl, pH 7.5, 10 mM MgCl$_2$) for at least 4 h at 37 °C, and subsequently treated with 400 µg Proteinase K (Sigma, P2308; diluted in 10 mM Tris-HCl, pH 7.5) for 30 min at 50 °C. Cells were resuspended in 50 mM Tris-HCl, pH 8.0, sonicated and diluted 1:20 with 50 mM Tris-

HCl, pH 8.0 containing 0.5 μM SYTOX Green (Invitrogen, S7020) and measured using a MACSquant Analyzer Flow Cytometer (Milteny Biotec). Data were analyzed and plotted using FlowJo (version 10.5.3) (FlowJo LLC) as previously described[8]. U2OS were collected by scraping, pelleted, and washed twice with PBS. The pellet was resuspended in 30 μl PBS and 500 μl of cold (−20 °C) methanol were added dropwise while slowly vortexing. Samples were stored at −20 °C. Cells were then washed twice with PBS + 0.01% Triton X-100, resuspended in 300 μl PI buffer (PBS, 0.01% Triton X-100, 10 μg/ml RNase A, 25 μg/ml propidium iodide (P3566, Invitrogen)) and incubated for 1 h at 37 °C. Samples were then filtered with 40 μm filters and measured using a MACSquant Analyzer Flow Cytometer (Milteny Biotec). Data were analyzed and plotted using FlowJo (version 10.5.3) (FlowJo LLC).

## SDS-PAGE and Western blot
For yeast cells, approximately $2 \times 10^7$ cells (1 OD) were harvested and frozen in liquid nitrogen. The pellet was then resuspended in 1 ml of cold water, 150 μl 1.85 M NaOH, and 7.5% β-mercaptoethanol and incubated at 4 °C for 15 min. 150 μl of 55% trichloroacetic acid (TCA, Roth, 8789.2) were added and samples incubated for 10 min on ice before centrifugation. The pellet was resuspended in 50 μl HU buffer (8 M urea, 5% SDS, 200 mM Tris-HCl, pH 6.8, 1.5% DTT, traces of bromophenol blue) and heated for 10 min at 65 °C. U2OS cells were collected by scraping, pelleted, and resuspended in 40 μl Laemmli buffer (2× concentrated Laemmli buffer: 0.18 M SDS, 0.5 M β-mercaptoethanol, 0.16 M Tris-HCl pH 6.8, 20% (v/v) glycerol, traces of bromophenol blue) and heated at 95 °C for 10 min.

Samples were run on NuPAGE 4–12% Bis-Tris acrylamide gels (Invitrogen, NP0323) using MOPS buffer (50 mM MOPS, 50 mM Tris base, 0.1% SDS, 1 mM EDTA) for 1 h at 180 V, unless to resolve phospho-shifts or for γH2AX blots. To resolve Sae2-9Myc, Sae2-3Flag and Dna2-9Myc phospho-shift, samples were run on 10% acrylamide gels in SDS buffer (25 mM Tris base, 192 mM glycine, 0.1% SDS) for at least 3 h at 170 V at 4 °C. For the CDK-dependent shift of Sld2, samples were run on NuPAGE 12% Bis-Tris acrylamide gels (Invitrogen, NP0343) for 3 h at 185 V at 4 °C[8]. For γH2AX blots, samples were run on 12% Bis-Tris acrylamide gels (Invitrogen, NP0343) using MES buffer (50 mM MES, 50 mM Tris base, 0.1% SDS, 1 mM EDTA) for 40 min at 200 V.

For western blot, proteins were transferred to a nitrocellulose membrane using transfer buffer (48 mM Tris base, 39 mM glycine, 0.0375% SDS, 20% methanol) at 4 °C for 90 min at 90 V. Membranes were washed for 5 min with western wash buffer (0.2% NP-40 in TBS) and incubated with primary antibody overnight at 4 °C while shaking. Membranes were washed twice for 5 min with western wash buffer and incubated with secondary antibody for 2 h at room temperature. After incubation membranes were washed 6 times for 5 min and detection was achieved using Pierce ECL reagents (Thermo Fisher Scientific, 32106) and a LAS-300 CCD camera system (Fujifilm). Antibodies for western blot were diluted in superblotto (2.5 % skim milk powder, 0.5% BSA, 0.5% NP-40, 0.1% Tween in TBS). A list of antibodies used in this study is provided in Supplementary Table 2. Western blot quantification was performed using Fiji-ImageJ (v.2.14.0/ 1.54 f). Images of uncropped western blots are presented in the Source Data file.

## λ-phosphatase treatment
100 ODs (1 OD corresponds to around $2 \times 10^7$ cells) of nocodazole-arrested cells were collected and washed twice with sorbitol buffer (25 mM HEPES-KOH, pH 7.6 1 M sorbitol) and resuspended in 1 volume of lysis buffer (50 mM HEPES-KOH, pH 7.6, 100 mM NaCl, 0.1% NP-40, 10% glycerol, 2 mM β-mercaptoethanol, protease inhibitors (400 μM PMSF, 4 μM aprotinin, 4 mM benzamidine, 400 μM leupeptin, 300 μM pepstatin A)), and frozen by dripping in liquid nitrogen. Frozen pellets were lysed using Cryo Mill (Spex SamplePrep 6870) with 6 cycles at a rate of 15 cps (cycles per second) for 2 min each. Protein concentration

in the extracts was measured using Bradford, and a volume corresponding to 500 μg of proteins per reaction was supplemented with 2 mM $MnCl_2$. Samples were either mock-treated or treated for 30 min with 4000 U of λ-phosphatase (New England Biolabs, P0753L) at 30 °C while shaking at 1200 rpm. Reactions were stopped by adding an equal volume of 2× concentrated Laemmli, followed by incubation for 5 min at 95 °C.

## Gene conversion assay 1 – recombination/gene conversion (qPCR)
A culture of cells (derivatives of YCL88 and YCL94) grown to stationary phase in YPR (supplemented with 40 mg/l adenine) was used to inoculate cells in YPR. At $OD_{600}$ of 0.5 cells were arrested in G1 or in M-phase. Subsequently, 2% galactose was added to the medium for DSB induction by pGAL-HO. For each time point, $4 \times 10^7$ cells (2 ODs) were collected (samples for DNA content analysis and western blot were collected as well). Cells were centrifuged and the pellet frozen in liquid nitrogen. Pellets were used for genomic DNA extraction using the Masterpure Yeast DNA Purification Kit (Lucigen, MPY80200) following manufacturer's instruction.

To measure the recombination product, around 20 ng of genomic DNA were used for qPCR using a control primer pair annealing on chromosome XV at a final concentration of 0.06 μM (to measure and normalize to the total amount of DNA) and a primer pair able to give a product only upon recombination-mediated repair at a final concentration of 0.6 μM (see Supplementary Data 3 for a list of primers used for qPCR experiments in this study). A calibration curve was generated for each primer pair using a template DNA containing the recombination product to provide concentration values, used to determine the relative recombination rate, via the ratio between the signal of the "recombination" primer pair and the signal of the "chr. XV" primer pair. Within the same qPCR plate, each sample was measured in three technical replicates. qPCRs were run on a Light-Cycler480 System (Roche) using KAPA SYBR FAST 2× qPCR Master Mix (KAPA Biosystems, London) for the master mix with an annealing temperature set to 59 °C.

## Gene conversion assay 2 – cell survival (colony-forming unit, cfu)
Cells (YCL88, YCL94, and derivatives) were grown in YPR (supplemented with 40 mg/l adenine). Cultures were then diluted to an $OD_{600}$ of 0.5 and 5-fold serial dilutions were prepared and spotted on YPD (as control) and YPGal (for induction of the pGAL-HO endonuclease) plates. Plates were left to grow for 2/3 days at 30 °C.

## Chromatin immunoprecipitation (ChIP)
Chromatin immunoprecipitation (ChIP) was performed as previously described[51,68]. Cells were lysed in lysis buffer (50 mM HEPES-KOH, pH 7.5, 150 mM NaCl, 1 mM EDTA, 1% Triton X-100, 0.1% Na-deoxycholate, 0.1% SDS) with zirconia beads using a bead beater (Retsch). Chromatin was sonicated to get fragments around 200-500 bp using a Bioruptor (Diagenode) in ice-cooled water. Cell lysates were centrifuged at 4 °C at 6150 g and diluted 1:1 with lysis buffer. 1% of the extract was taken as input sample and/or total DNA sample. RPA and single-stranded DNA were immunoprecipitated using an anti-RFA antibody (Agrisera, AS07214) for 2 h. Subsequently, protein A dynabeads (Invitrogen, 10002D) were added to the mixture for 30 min. Beads were first washed three times with lysis buffer, second with lysis buffer with additional 500 mM NaCl, third with wash buffer (10 mM Tris-HCl, pH 8.0, 0.25 M LiCl, 1 mM EDTA, 0.5% NP-40, 0.5% Na-deoxycholate) and last with TE buffer (pH 8.0). Immunoprecipitated complexes were eluted using 1% SDS and proteins were degraded by addition of 1 μg/μl proteinase K for 3 h at 42 °C and crosslinks were reversed (8 h at 65 °C). DNA was then purified using phenol-chloroform extraction. Residual phenol-chloroform was removed using phase-lock gel tubes (Quantabio) and DNA was precipitated with ethanol. DNA concentration was

determined using Qubit 3.0 fluorometer (Invitrogen) and Qubit dsDNA HS assay kit (Invitrogen).

For RPA-ChIP-qPCR experiments, resection was measured as previously described[51], using primer pairs at the indicated positions (up to 20 kb downstream the DSB; see Supplementary Data 3 for a list of primers used for qPCR experiments in this study).

## Strand-specific NGS

Strand-specific NGS was performed as previously reported[68]. NGS libraries were prepared starting from 1 to 1.5 ng of DNA using Accel-NGS- 1 S Plus Library Kit (Swift BioScience) following manufacturer's instructions and 10–12 cycles for library amplification. For clean-up steps, SPRIselect beads were used. To assess the size distribution and the concentration of libraries, high sensitivity DNA Chip with Bioanalyzer 2100 (Agilent Genomics) was used. DNA was paired-end sequenced with 75 cycles per read on an Illumina NextSeq 500 sequencer or with 60 cycles per read on an Illumina Novaseq 6000 sequencer (NGS facility Max Planck Institute of Biochemistry).

## Quantification and NGS analysis

Sequencing reads were mapped to the *S. cerevisiae* genome using Bowtie2 (v2.4.2) with the parameters --no-discordant --no-mixed --local. The *S. cerevisiae* LYZE00000000.1 genome assembly[109] was used for mapping after removing the sequences of the *HML* and *HMR* loci, which are deleted in all the strains used for NGS experiments. SAMtools (v1.9) was used to filter out multiple mapping reads and select reads with alignment quality higher than 30 (MAPQ > 30). Analyses for every dataset shown were performed using the R package (RStudio Version 2022.12.0, R version 4.2.2).

Plots representing total DNA signal over the HO-induced break (Fig. 2d, Supplementary Fig. 5d) were generated by first subsampling sequencing reads to the same sequencing depth per sample. The coverage for each time point after DSB induction was then divided by the coverage of the uninduced sample. The normalized coverage of a 20 kb window around the DSB (divided in 2 kb bins) for each time point was plotted with ggplot2.

RPA enrichment was calculated by normalizing the RPA sequencing coverage at the DSB to unspecific RPA peaks that were found throughout the genome in all experiments. These peaks were selected using the IRanges package (v2.32.0) function "slice" with the lower value set to 6× the median coverage. Only peaks with a width between 100 and 500 bp and overlapping between different samples and replicates of the same type of experiment (arrest in G1-phase or in M phase of the cell cycle) were retained. Normalized forward and reverse coverage was then plotted over a 25 kb window on both sides of the DSB using R base plotting functions.

## Resection assay

Resection assays were performed as previously described[81]. Briefly, cells were grown exponentially in YPD at 30 °C. Cells were arrested in M-phase by adding 1% DMSO and 20 µg/ml Nocodazole (AbMole, M3194) and further cultured for 1.5 generation times. AID-tagged proteins were depleted by addition of 1 mM IAA (abcam, ab146403) for 1.5 h. For DSB formation, 2 µM β-estradiol (Sigma, E8875) was added to the culture and cells were harvested at the different time points by centrifugation, washed with TE (10 mM Tris, 1 mM EDTA, pH 8.0) containing 0.1% sodium azide, and cell pellets stored at −20 °C. Genomic DNA was extracted using the MasterPure Yeast DNA Purification Kit (Lucigen), including RNase treatment. Genomic DNA extracts were diluted to 2.25 ng/µl in CutSmart buffer (New England Biolabs) and 100–150 ng were digested with 5 U RsaI or MseI (New England Biolabs), respectively. DSB formation was evaluated at the *MAT*a HO cut site and resection was evaluated past RsaI or MseI sites located 120 bp upstream or 98 bp, 640 bp, 2.5 kb or 5 kb downstream the DSB by qPCR. Per sample, we analyzed triplicates each containing

10 ng (un-)digested genomic DNA, 300 nM of each primer (see Supplementary Data 3 for a list of primers used for qPCR experiments in this study), and 1× SsoAdvanced Universal SYBR Green Supermix (Biorad) in a total volume of 10 µl on a CFX384 Real-Time System (Biorad) with 10 min initial denaturation at 95 °C and 40 cycles of 10 s denaturation at 95 °C and 1 min annealing and extension at 58 °C.

qPCR data were analyzed and visualized with the statistical software R (version 4.3.1).

## Phospho-proteomics

**Cell pellets preparation.** Cells from strains YBP388, YLP100, and YLP101 were grown in YPD to an $OD_{600}$ of 0.5 and arrested in the M-phase of the cell cycle with nocodazole. 200 ODs (1 OD corresponds to around $2 \times 10^7$ cells) of cells were then centrifuged, and the pellet washed twice in PBS 1×. Pellets were then stored at −80 °C until further use.

**Sample preparation.** The cell pellets were incubated with 6 ml of preheated SDC buffer containing 1% sodium deoxycholate (SDC, Sigma–Aldrich), 40 mM 2-cloroacetamide (CAA, Sigma–Aldrich), 10 mM tris(2-carboxyethyl)phosphine (TCEP; Thermo Fisher Scientific) and 100 mM Tris, pH 8.0. After incubation for 2 min at 95 °C, the samples were ultrasonicated for 2 min with 0.5 seconds pulse (50% intensity) and 0.2 s pause (Sonopuls, Bandelin). Incubation and ultrasonication was repeated for a second time. After a final incubation for 2 min at 95 °C, 1/6 of the sample was diluted 1:2 with MS grade water (VWR). Proteins were digested overnight at 37 °C with 50 µg trypsin (Promega). The solution of peptides was then acidified with trifluoroacetic acid (Merck) to a final concentration of 1%, followed by desalting via Sep-Pak C18 5cc vacuum cartridges (Waters). The cartridge was washed twice with 1 ml of 100% methanol, twice with 1 ml of 0.1% FA in 80% ACN and twice with 1 ml of 0.1% FA in water prior to sample loading. After loading the acidified sample, the cartridge was washed twice with 1 ml of 0.1% FA in water. Elution was done with 2 × 1 ml of 0.1% FA in 80% ACN.

**Phospho-peptide enrichment.** To 1 ml of the desalted peptides, 400 µl isopropanol and 100 µl of enrichment buffer (48% TFA, 8 mM K2HPO4) was added and mixed at room temperature at 1500 rpm. Then, $TiO_2$ beads (10 mg) in EP loading buffer (80% ACN/6% TFA) were subsequently added and incubated at 37 °C for 5 min at 2000 rpm. Beads were subsequently pelleted by centrifugation for 1 min at 3500 g, and the supernatant (containing non-phospho-peptides) was aspirated. Beads were suspended in wash buffer (60% ACN, 1% TFA) and transferred to clean vials, and washed a further four times with 1 ml wash buffer. After the final wash, beads were suspended in 150 µl transfer buffer (60% isopropanol, 0.1% TFA) and transferred onto the top of a C8 StageTip, and centrifuged for 3–5 min at 500 g or until no liquid remained on StageTip. Bound phospho-peptides were eluted 2× with 30 µl elution buffer (40% ACN, 20% NH4OH (25%, HPLC grade)), and collected by centrifugation into clean PCR tubes. Samples were concentrated in a SpeedVac for 15 min at 45 °C.

**LC–MS/MS data acquisition.** Peptides were loaded onto a 30-cm column (inner diameter: 75 microns; packed in-house with ReproSil-Pur C18-AQ 1.9-micron beads, Dr. Maisch GmbH) via the autosampler of the Thermo Easy-nLC 1000 (Thermo Fisher Scientific) at 60 °C. Using the nanoelectrospray interface, eluting peptides were directly sprayed onto the benchtop Orbitrap mass spectrometer Q Exactive HF (Thermo Fisher Scientific). Peptides were loaded in buffer A (0.1% (v/v) formic acid) at 250 nl/min and percentage of buffer B (80% acetonitrile, 0.1% formic acid) was ramped to 30% over 120 min followed by a ramp to 60% over 10 min then 95% over the next 5 min and maintained at 95% for another 5 min. The mass spectrometer was operated in a data-dependent mode with survey scans from 300 to 1750 m/z

(resolution of 60000 at m/z = 200), and up to 12 of the top precursors were selected and fragmented using higher energy collisional dissociation (HCD with a normalized collision energy of value of 28). The MS2 spectra were recorded at a resolution of 15000 (at m/z = 200). AGC target for MS and MS2 scans were set to 3E6 and 1E5 respectively within a maximum injection time of 20 ms for MS1 and 50 ms for MS2 scans. Dynamic exclusion was set to 16 ms.

**Data analysis 1.** Raw data were processed using the MaxQuant computational platform with standard settings applied. Shortly, the peak list was searched against the reviewed Uniprot yeast proteome database (proteome ID: UP000002311; downloaded in 2019) with an allowed precursor mass deviation of 4.5 ppm and an allowed fragment mass deviation of 20 ppm. MaxQuant by default enables individual peptide mass tolerances, which was used in the search. Trypsin/P was set as protease. Cysteine carbamidomethylation was set as static modification, and methionine oxidation, N-terminal acetylation, deamidation and phosphorylation as variable modifications. The identifications were filtered at 1% FDR at the proteinGroups and peptide level.

**Data analysis 2 - Perseus analysis.** Data were analyzed using Perseus V1.6.15.0[110] as summarized in Supplementary Fig. 1d. The enriched phospho-peptides and the total proteome were both analyzed. For the total proteome, the same analysis was performed in parallel to compare LFQ intensities of: (I) WT and *bob1-1 dbf4Δ*, (II) *bob1-1* and *bob1-1 dbf4Δ* cells. Missing values were imputed using default settings (width: 0.3, down-shift: 1.8, mode: separately for each column). ANOVA (analysis of variance) was performed with permutation-based FDR (false discovery rate) = 0.05. Proteins that were found to be significantly changing were highlighted (based on gene names) and those changing in both total proteome comparisons were removed from the phospho-peptide analysis (following). It should be noted that proteins for which phospho-peptides are identified, but that are not detected in the total proteome in any strain, are not sensitive to this analysis. For the analysis of the phospho-peptides, only those with a localization probability >0.75 (class-I peptides[111]); were retained for the analysis. Missing values were imputed using default settings (width: 0.3, down-shift: 1.8, mode: separately for each column). ANOVA was performed with permutation-based FDR = 0.05 and ANOVA significant hits were maintained. The z-score was then calculated (at this stage 3981 phospho-peptides were present). Proteins changing at total proteome level (as described above) were then removed from the analysis (leaving 3587 phospho-peptides). A hierarchical clustering of the phospho-peptides was performed for rows and columns (distance: Euclidean, linkage: average, constraint: none, preprocessing with K-means: no; heat-map shown in Fig. 1d) and six clusters were defined. The sequence motif logo (Fig. 1e) was generated using WebLogo with default settings (https://weblogo.berkeley.edu/logo.cgi[112]); Gene Ontology (GO) analysis of Biological Processes (BP) was performed using Panther (https://geneontology.org[113,114]), with default settings (Supplementary Data 1), using Perseus' list 'Gene names' of the DDK cluster as input. For filtering based on GOBP terms in Perseus, the annotation *mainAnnot.saccharomyces_cerevisiae_s288c* was used. Annotation was downloaded (following the path: OldReleases, 2019_01, OrganismSpecific, s) from https://datashare.biochem.mpg.de/s/qe1IqcKbz2j2Ruf?path=%2FPerseusAnnotation. A list of the sample names for the deposited files is provided in Supplementary Table 3.

### Single-strand annealing assay
Strains carrying a 25 kb spacer between a galactose-inducible HO cut site and repair locus (derivatives of the strain YMV80[66]) were grown in YPR (supplemented with 40 mg/l adenine). Cultures were then diluted to an $OD_{600}$ of 0.5 and 5-fold serial dilutions were prepared and spotted on YPD and YPGal plates. Plates were left to grow for 2/3 days

at 30 °C. Note that cells were deleted of *RAD51* in order to block DSB repair via gene conversion.

### Hairpin recombination assay (fluctuation analysis)
The rate of Lys+ recombinants (based on the system generated in[78]) was calculated by performing a fluctuation analysis starting from 7–8 independent colonies per strain per replicate. Single colonies were inoculated in 5 ml YPD and grown at 30 °C for 3 days. Cells were then collected, washed and dilutions were plated on YPD plates (as a control of the number of cells and viability) and -Lys plates (to count recombinants). Cells were plated from dilutions that were tested to give a countable number of colonies (between ~20 and 200). The recombination rate was then calculated using the FALCOR tool ([115]; https://lianglab.brocku.ca/FALCOR/), using a MSS Maximum likelihood approach.

### Human cell culture
Human osteosarcoma U2OS cells were cultured in DMEM Glutamax (61965-026, Gibco) supplemented with 10% fetal bovine serum (FBS, 10500-064, Gibco) and 100 U/ml Penicillin, 0.1 mg/ml Streptomycin (P06-07100, PAN Biotech) in an atmosphere containing 5% $CO_2$ at 37 °C.

### Immunofluorescence and imaging of fixed samples
For immunofluorescence experiments, per condition, 2 ml of U2OS cells at a density of around $1.5 \times 10^5$ cells/ml were seeded in a 6-well plate containing glass coverslips coated with Poly-L-lysine solution (P4707, Sigma). In parallel, per condition, 1 ml of the same batch of cells was seeded on a 12-well plate (in duplicates) for subsequent collection of samples for DNA content analysis and western blot. The next day, the medium was exchanged with medium containing 0.05 µg/ml nocodazole. After 16 h, cells were either mock-treated or treated for 6 h with 20 µM XL413 (SML1401, Merck) to inhibit DDK. Subsequently, cells were either mock-treated or treated for 2 h with 200 µg/ml zeocin (R25001, Invitrogen) for DNA damage induction. Samples for DNA content measurement and western blot were collected and processed as reported in sections "Flow cytometry" and "SDS-PAGE and western blot", respectively. Cells for immunofluorescence were pre-extracted and fixed as described below. On ice, cells were washed with ice-cold PBS and treated for 10 min with ice-cold extraction buffer 1 (10 mM Pipes, pH 7.0, 100 mM NaCl, 300 mM sucrose, 3 mM MgCl2, 1 mM EGTA, 0.5% Triton X-100), washed with ice-cold PBS 1× and treated for 10 min with ice-cold extraction buffer 2 (19 mM Tris-HCl, pH 7.5, 10 mM MgCl2, 1% Tween-40) as previously described[116]. Cells were then washed with PBS, fixed using 4% buffered paraformaldehyde (043368.9 M, Thermo Fisher) at room temperature for 20 min, and washed with PBS.

Cells were then incubated for 1 h at 4 °C in permeabilization/blocking solution (PBS + 0.3% Triton X-100, 1% DMSO, 5% normal goat serum (P30-1001, PAN Biotech)). Incubation with anti-RPA70 (ab79398, Abcam, 1:1000 in permeabilization/blocking solution) was performed overnight at 4 °C. Coverslips were washed three times for 5 min with PBS. Secondary antibody goat anti-rabbit Alexa Fluor 488 (A-11008, Invitrogen, 1:500 in permeabilization/blocking solution) incubation was performed at room temperature for 45 min together with DAPI (0.5 µg/mL, D9542, Merck) to stain the DNA. Following three washes of 5 min with PBS, coverslips were briefly washed with distilled water, the excess of liquid was dried and coverslips were mounted with fluoromount-G mounting medium (00-4958-02, Invitrogen) on glass slides (2951-001, Thermo Fisher).

Microscopy was performed using an Evos FL Auto 2 (Invitrogen) with a 60× oil-immersion objective. Acquired images were analyzed with Cell Profiler (v 4.2.5). To specifically count RPA foci in mitotic nuclei, a step of selection of the mitotic nuclei was performed using the "Identify objects manually" option in Cell Profiler (v 4.2.5) based on the

DAPI staining, and used as a mask for the images with the RPA foci. RPA foci were then counted using Cell Profiler (v 4.2.5) with manual supervision of the counting.

## DDK purification

YSDK8 cells were grown in 6 liters of YPR at 30 °C, and overexpression of both CBP-Dbf4 and Cdc7 was induced via addition of 2% galactose to log-phase cells, for 8 h at 30 °C. Cells were harvested and washed first with sorbitol buffer (25 mM HEPES-KOH, pH 7.6, 1 M sorbitol) and then with buffer I (25 mM HEPES-KOH, pH 7.6, 0.03 mM NP-40, 5% glycerol, 400 mM NaCl). Pellets were then resuspended in 1 volume lysis Buffer (25 mM HEPES-KOH, pH 7.6, 0.03 mM NP-40, 5% glycerol, 400 mM NaCl, 2 mM β-mercaptoethanol, supplemented with protease inhibitors (400 μM PMSF, 4 μM aprotinin, 4 mM benzamidine, 400 μM leupeptin, 300 μM pepstatin A, 100 nM okadaic acid)) and snap frozen by dripping in liquid nitrogen. Frozen pellets were lysed using Cryo Mill (Spex SamplePrep 6870) with 6 cycles at a rate of 15 cps (cycles per second) for 2 min each. The thawed powder was centrifuged at 250000 g for 1 h at 4 °C (in a WX Ultra 90 centrifuge (Sorvall) using a T-865 rotor (Thermo Scientific)) and the clear upper phase was recovered. 2 mM CaCl$_2$ was then added to the suspension together with 1.5 ml bed volume calmodulin affinity resin (Agilent, 214303) and stirred for 3 h at 4 °C. Beads were then recovered and washed 5 times with 3 ml wash buffer (lysis buffer supplemented with 2 mM CaCl$_2$, 0.1 mM EGTA, 0.1 mM EDTA), followed by 5 washes with 4.5 ml of ATP-wash buffer (25 mM HEPES-KOH, pH 7.6, 200 mM NaCl, 50 mM KCl, 10 mM magnesium acetate, 2 mM ATP, 5% glycerol, 0.03 mM NP-40, 2 mM β-mercaptoethanol, 0.1 mM EDTA, 0.1 mM EGTA, 2 mM CaCl$_2$) for removal of an otherwise co-purifying chaperone. After the ATP wash, beads were washed 10 times with 4.5 ml of wash buffer and eluted with 2 ml of elution buffer (lysis buffer supplemented with 2 mM EGTA, 1 mM EDTA) 10 times. Fractions containing DDK were then concentrated using Amicon 50 KDa concentrator (Merck, UFC505024) and snap frozen in liquid nitrogen.

## CDK2/CycA purification

CDK was purified from *E.coli* BL21 pRIL as previously described[15]. Briefly, human GST-CDK2 and bovine CycA-6His-ΔN170 were expressed separately with 1 mM IPTG overnight at 20 °C. Pellets were then resuspended in lysis buffer 1 (20 mM HEPES-KOH, pH 7.6, 300 mM NaCl, 5 mM β-mercaptoethanol, 0.01% NP-40, cOmplete protease inhibitor (Roche, 04693132001)) and lysed using an Avestin homogenizer (Avestin) with three rounds at 1000 bar, and after lysis supplemented with 1 mM PMSF. After centrifugation, the clear upper phases of each construct-expression were pooled and stirred at 4 °C for 1 h for CDK2-CycA complex formation. Glutathione agarose beads were added to the mixture and incubated for 2 h at 4 °C. After wash with lysis buffer 1 and with lysis buffer 2 (as lysis buffer 1 but with 150 mM NaCl) CDK2-CycA was eluted incubating the beads with 250 U GST-PreScission protease (core facility Max Planck Institute of Biochemistry). Eluate was then supplemented with 6 mM imidazole and NaCl concentration brought to 300 mM before Ni-NTA (Qiagen, 30210) pulldown. Elution was then achieved with elution buffer (150 mM NaCl, 20 mM HEPES-KOH, pH 7.6, 5 mM β-mercaptoethanol, 0.01% NP-40, 250 mM imidazole, 5% glycerol). CDK2-CycA was then dialyzed in dialysis buffer (20 mM HEPES-KOH, pH 7.6, 150 mM NaCl, 5 mM β-mercaptoethanol, 5% glycerol) and aliquots were then prepared and snap frozen in liquid nitrogen.

## MRX and Sae2 purification

The Mre11-Rad50-Xrs2 proteins were expressed as a complex in *Spodoptera frugiperda* 9 (*Sf*9) cells. The Mre11 construct contained a His-tag at the C-terminus of Mre11, and a FLAG-tag at the C-terminus of Xrs2. The proteins were then purified using affinity chromatography using NiNTA resin (Qiagen) followed by anti-FLAG M2 affinity resin (Sigma)[117]. Non-phosphorylated Sae2 (MBP-tagged at the N-terminus and his-tagged at the C-terminus) was similarly prepared in *Sf*9 cells[46]. The soluble extract was prepared without phosphatase inhibitors and construct was first bound to amylose resin (New England Biolabs). After elution in MBP elution buffer (50 mM Tris-Cl pH 7.5, 5 mM β-mercaptoethanol, 10% glycerol, 10 mM maltose, 300 mM sodium chloride, 1 mM magnesium chloride), 20'000 units of λ-phosphatase (New England Biolabs) was added to the protein obtained from 1.6 L *Sf*9 cell culture (approximately 1 mg). The reaction was incubated for 30 min at 30 °C. PreScission protease was added (1:8, w:w) to cleave off the MBP tag, and the solution was incubated for 2 h at 4 °C. The C-terminally His-tagged Sae2 was bound to NiNTA resin (Qiagen), eluted with in a buffer with 300 mM imidazole, and dialyzed into storage buffer (50 mM Tris-HCl pH 7.5, 5 mM β-mercaptoethanol, 10% glycerol, 300 mM sodium chloride), and stored at −80 °C.

## In vitro kinase assays

Radioactive in vitro phosphorylation reactions were performed in 20 μl volume in 25 mM Tris-acetate, 10 mM magnesium acetate, 100 mM NaCl, and 0.25 mg/ml Avidin (Sigma). Recombinant non-phosphorylated Sae2 (800 ng) and 300 nM of recombinant CDK2-CycA or DDK kinases were combined, the reactions were initiated by the addition of 0.1 mM ATP and 37 kBq of [γ-$^{32}$P]ATP (Perkin Elmer), and incubated for 30 min at 30 °C. The reactions were stopped by adding 5 μl of SDS buffer (50 mM Tris-HCl, pH 6.8, 1.6% sodium dodecyl sulfate, 10% glycerol, 0.1 M DTT, 0.01% bromophenol blue), and incubated for 5 min at 95 °C. The samples were separated by SDS-PAGE, first stained with Instant Blue (LucernaChem) and subsequently dried on 3MM paper (Whatman), exposed to storage phosphor screens (GE Healthcare), and scanned by a Typhoon 9500 phosphorimager (GE Healthcare). When phosphorylated Sae2 was used for biochemical assays, the reactions were carried out similarly, but without radioactive ATP and with 2 mM unlabelled ATP. Additionally, BSA (0.25 mg/ml) was used instead of Avidin. Following the 30 min kinase reaction time, Sae2 was diluted into the nuclease assays. Mock-treated Sae2 was handled in the same way but without the respective kinase.

## Nuclease assay

Nuclease assays (15 μl volume) were carried out with $^{32}$P-labeled oligonucleotide-based DNA substrate blocked at all ends with streptavidin (1 nM molecules, oligonucleotides, 210: GTAAGTGCCGCGGTGCGGG TGCCAGGGCGTGCCCTTGGGCTCCCCGGGCGCGTACTCCACCTCATG CATC and 211: GATGCATGAGGTGGAGTACGCGCCCGGGGAGCCCAA GGGCACGCCCTGGCACCCGCACCGCGGCACTTAC)[25], in a buffer containing 25 mM Tris-acetate, pH 7.5, 1 mM dithiothreitol, 5 mM magnesium acetate, 1 mM manganese acetate, 1 mM ATP, 80 U/ml pyruvate kinase (Sigma), 1 mM phosphoenolpyruvate, 0.25 mg/ml bovine serum albumin (New England Biolabs). 30 nM streptavidin (Sigma), was added to the substrate containing the reaction mixture and incubated for 5 min at room temperature. Recombinant proteins were then added on ice and incubated for 30 min at 30 °C. Reactions were terminated and the DNA products were separated by denaturing gel electrophoresis and detected by autoradiography[46] and the data were quantitated.

## Statistical analysis

Reported *p*-values were calculated using a two-tailed unpaired *t*-test with the Graphpad *t*-test calculator web tool (https://www.graphpad.com/quickcalcs/ttest1.cfm; Graphpad), with exception for the *p*-values reported in Fig. 3g, h, Fig. 4e, f, Supplementary Fig. 3e-f which were calculated using a two-tailed unpaired *t*-test with R and the *p*-values reported in Fig. 2f which were calculated using a two-tailed Mann-Whitney test (Prism 8, Graphpad). *p*-values reported in Supplementary Data 1 were calculated selecting Fisher's exact test in the GOBP analysis with Panther (https://geneontology.org).

**Reporting summary**

Further information on research design is available in the Nature Portfolio Reporting Summary linked to this article.

## Data availability

The ChIP-seq data generated in this study have been deposited in NCBI's Gene Expression Omnibus under GEO Series accession number GSE233549. The mass spectrometry data generated in this study have been deposited in the ProteomeXchange Consortium via the PRIDE partner repository under accession number PXD042607. Source data are provided with this paper.

## Code availability

The scripts used in this study will be made available upon request to the corresponding author.

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

## Acknowledgements

We thank Rinho Kim (MPIB NGS facility) for NGS, Sandra Mitzkus, Uschi Schkölziger, and Denise Marie Roggan for technical assistance, John Diffley, James E. Haber, Claudio Lademann, Ralf Jungmann, Stephanie Panier, Michael Resnick and Karl-Uwe Reusswig for strains, cell lines and plasmids, Reinhard Fässler for access to microscopes, members of the Pfander lab for stimulating discussion and critical reading of the manuscript. This work was supported by funding of the Max Planck Society, the German Aerospace Center (DLR), and TU Dortmund University, grants by the Deutsche Forschungsgemeinschaft (DFG, German Research Foundation): 466479039 and 445098914 (to BP), grants by the Swiss National Science Foundation (SNSF) SNSF 310030_207588, SNSF 310030_205199 and European Research Council (ERC) ERC 101018257 (to PC) and a grant by the National Institutes of Health (NIH) NIH R35 GM126997 (to LS).

## Author contributions

L.G., P.C., and B.P. conceived and designed research. L.G. performed all experiments (with following exceptions), M.P. performed experiments in Fig. 2d, e, Supplementary Fig. 2a, Supplementary Fig. 2d, Fig. 5c, Supplementary Fig. 5d and analyzed NGS data, R.G. performed experiments in Fig. 3g, h, Fig. 4e, f Supplementary Fig. 3d–g, Supplementary Fig. 4g, h, E.C. performed experiments in Fig. 3a–c, Supplementary Fig. 3b, J.H. performed experiment in Fig. 3l, B.S. performed mass-spectrometry run in Fig. 1d, M.D.P. contributed with unpublished data. All authors analyzed the data. L.G. and B.P. wrote the paper, all authors contributed to manuscript writing. B.P., P.C., and L.S. acquired funding.

## Funding

## Competing interests

The authors declare no competing interests.
