## [Peer Review File · Nature Communications]

Dbf4-dependent kinase promotes cell-cycle controlled resection of DNA double-strand breaks and repair by homologous recombinationREVIEWER COMMENTS

Reviewer #1 (Remarks to the Author):

In the present study, Galanti et al. focused on DDK-dependent phosphorylation and its role in the DSB response. First, the authors successfully distinguished the role of DDK in DNA replication and DSB response using a *bob1-1* mutant that bypasses the DDK functional requirement. This experiment is very important, because although it was previously known that DDK is involved in homologous recombination among DSB repairs, it was not known whether the function of DDK is mediated by DNA replication, as DDK is essential for DNA replication, and this study is the first demonstration of DDK's direct involvement in repair factors in mitosis.

Of course, there is still much that is not known. For example, the detailed role of phosphorylation, including protein-protein interactions, remains unexplored. The authors have determined that the requirement for DDK-dependent phosphorylation for homologous recombination repair, but the contribution of these phosphorylation to HR under physiological conditions, including the time course of the phosphorylation in the cell, or the correlation between the actual timing of phosphorylation in the physiological cell cycle to HR activity, is still unresolved.

However, the broad information of and molecular detail for DDK-mediated protein regulation in cells and possible collaboration between Sae2 and Dna2 (Figure 4), mechanistic insight on how resection is regulated (Figure 5), combined with phosphoproteomics, have been described. These are really informative to the research community for further understanding of spatio-temporal regulation of DSB repair in cells.

Overall, yeast genetic experiments are very carefully done, and interpretation of the data is very modest and deeply considered. Therefore this reviewer think that the manuscript is a good candidate for being published in this journal.

Major comment:

1. Figure 1b, Figure S1b: I could not find the experimental detail of HR survival assay. I assume that HO was induced under GAL promoter, and the template DNA was harbored as a ectopic plasmid. Better present in materials and method or as a supplement figure.

2. Figure 2f, The data show the effect of XL-413 on DDK inhibition on RPA foci formation in U2OS cells, where XL-413 inhibition was preceded by G2 arrest with a CDK inhibitor. This suggests that the experimental system itself is more focused on DDK inhibition. This point should be mentioned in the text.

An important information is also lacking. It has not been shown whether the amount of damage is similarly induced by gamma radiation or endogenously between these XL-413 inhibitory and non-inhibitory conditions. eg: γ -H2AX foci numbers should be stated to monitor DNA damage response. In fact, it is not clear whether γ -H2AX foci themselves are formed in the same way depending on the repair situation, so it would be better to describe them as they are. But is very important information for the readers.

3. In Figure 3a, it seems that when DDK and CDK is combined in the kinase assay the phosphor-incorporation is less as compared to the CDK only sample. Why is that? Better explain this somewhere.

Reviewer #2 (Remarks to the Author):

In this manuscript Galanti and co-workers describe the function of the CDC7/DBF4 kinase in the

regulation of DNA double-strand breaks resection and the consequences of impaired CDC7/DBF4 activity on homologous recombination in budding yeast.

The authors combine molecular genetics, proteomics, biochemical and functional approaches, showing that two nucleases, Sae2/MRN and DNA2, are the target of CDC7/DBF4 activity, regulating short- and long-range resection respectively. They provide substantial mechanistic insight on how CDC7 works together with CDK1 in determining the cell cycle dependent pathway choice in dsDNA breaks repair. Finally, the authors also provide some evidence that the role of CDC7 in resection is conserved in human cells.

The experiments performed are well-designed, consequential, and elegant; the data are clearly presented and fully supports the claims. Methodology is rigorous and of very high standard throughout; the manuscript is well written and sufficient detail is given in the M&M section.

This is an important work clarifies how two kinases, CDC7/DBF4 and CDK1, which together have a major role in driving cells through the cycle-cycle, commonly regulate HR, and it is a significant advance in the field of genome stability.

I only have a couple of minor observations:

- 1) the experiment which shows that CDC7 inhibition potentially affect resection in U2OS cells (Fig 2FG) is performed in cells also treated with a CDK1 inhibitor (RO-3306), which is likely to also affect efficiency of resection. This should be mentioned and discussed.
- 2) In Fig3 g-h is not clear if the differences in short-range resection between WT and dbf4-3aid are significant and if in these experimental conditions resection is defective or simply delayed. Can the effect of CDK1 inhibition be compared in the same assay?

Reviewer #3 (Remarks to the Author):

The study reported by Galanti et al in the current manuscript being considered for publication in Nature Communications makes several important contributions to the DNA repair field. Chief among these are the findings that: (i) DDK promotes Homologous Recombination following DNA double-strand breaks, through stimulation of end resection, (ii) DDK appears to do so through phosphorylation of key factors involved in resection initiation (Sae2) and resection elongation (Dna2), (iii) engineered DDK activation in G1 phase can result in limited resection at a cell cycle stage where this does not normally occur.

Overall, the work is presented in a clear fashion, with convincing results that back up the main claims made by the authors. In my view, the main value to the field is in uncovering mechanistic details of budding yeast DDK promotion of HR-directed repair of DNA DSBs. While some initial results, and previous cited reports, are consistent with conservation of this role for DDK in humans, this is not explored in a substantive way in the present manuscript.

I am supportive of publication, but the following points should be considered for a revised manuscript:

- The evidence supporting conservation in human cells of a DDK role in promoting HR-directed DNA DSB repair is very limited, consisting of only the data in Fig. 2F.
- Was there confirmation of M-phase arrest of the U2OS cells treated with RO-3306? While flow cytometry analysis was performed to confirm cell-cycle arrest for yeast cells in several instances, I saw no mention of this being carried out for the U2OS cells.

- Given that RO-3306 is a CDK1 inhibitor, it is important to factor dual inhibition into the interpretation of the DDK impairment results. Was consideration given to inhibiting DDK through another means in the U2OS cells (e.g. siRNA-mediated depletion)?

- For the sae2-14A strain, the lack of growth on even low CPT concentrations (Fig.3K) may also have been due to lower stability of the corresponding mutant Sae2 protein, similarly to what was noted by the authors in relation to the defect in its phosphorylation

- The authors indicate for their assessment of recombination-mediated repair (Fig. 5D), that cells were maintained in a state of G1 arrest throughout the experiment, however the flow cytometry profiles (Fig.S5E) seem to indicate some progression into S-phase for all the strains examined for the 5-9h timepoints

- Typos: Last paragraph of Results, instead of "We ruled out that these effects...", seems to me it should be "We confirmed that these effects..."; there are a couple of instances where the phrase "difficult to entangle" appears when it should be "difficult to disentangle"

Point-by-point response to reviewer comments to Galanti et al. “Dbf4-dependent kinase promotes cell-cycle controlled resection of DNA double-strand breaks and repair by homologous recombination”

Reviewer #1:

In the present study, Galanti et al. focused on DDK-dependent phosphorylation and its role in the DSB response. First, the authors successfully distinguished the role of DDK in DNA replication and DSB response using a bob1-1 mutant that bypasses the DDK functional requirement. This experiment is very important, because although it was previously known that DDK is involved in homologous recombination among DSB repairs, it was not known whether the function of DDK is mediated by DNA replication, as DDK is essential for DNA replication, and this study is the first demonstration of DDK's direct involvement in repair factors in mitosis.

Of course, there is still much that is not known. For example, the detailed role of phosphorylation, including protein-protein interactions, remains unexplored. The authors have determined that the requirement for DDK-dependent phosphorylation for homologous recombination repair, but the contribution of these phosphorylation to HR under physiological conditions, including the time course of the phosphorylation in the cell, or the correlation between the actual timing of phosphorylation in the physiological cell cycle to HR activity, is still unresolved.

However, the broad information of and molecular detail for DDK-mediated protein regulation in cells and possible collaboration between Sae2 and Dna2 (Figure 4), mechanistic insight on how resection is regulated (Figure 5), combined with phosphoproteomics, have been described. These are really informative to the research community for further understanding of spatio-temporal regulation of DSB repair in cells.

Overall, yeast genetic experiments are very carefully done, and interpretation of the data is very modest and deeply considered. Therefore this reviewer think that the manuscript is a good candidate for being published in this journal.

We thank this reviewer for his/her considerations and the careful assessment of our work. We very much appreciate the highly positive feedback.

Major comment:

1. Figure 1b, Figure S1b: I could not find the experimental detail of HR survival assay. I assume that HO was induced under GAL promoter, and the template DNA

was harbored as an ectopic plasmid. Better present in materials and method or as a supplement figure.

Apologies! We now provide an extensive experimental description of the survival assay in the materials and methods section and also modified the experimental scheme in Fig. 1b. Briefly, we use the same HR tester strain for both assays. Therefore, a site-specific DSB is induced by pGAL-HO and repair can take place using homologous DNA from an ectopic locus on the same chromosome.

2. Figure 2f, The data show the effect of XL-413 on DDK inhibition on RPA foci formation in U2OS cells, where XL-413 inhibition was preceded by G2 arrest with a CDK inhibitor. This suggests that the experimental system itself is more focused on DDK inhibition. This point should be mentioned in the text.

An important information is also lacking. It has not been shown whether the amount of damage is similarly induced by gamma radiation or endogenously between these XL-413 inhibitory and non-inhibitory conditions. eg: g-H2AX foci numbers should be stated to monitor DNA damage response. In fact, it is not clear whether g-H2AX foci themselves are formed in the same way depending on the repair situation, so it would be better to describe them as they are. But is very important information for the readers.

All 3 reviewers have raised the point that we used RO3306 for G2 arrest in the experiment shown in Fig. 2f. As RO3306 is a CDK1 inhibitor, this strategy may be considered sub-optimal given the involvement of CDK in phosphorylation of CtIP. We therefore replaced this experiment with an alternative experiment, where we arrested U2OS cells in metaphase using nocodazole in a state of high CDK1 and CDK2 activity (Fig. 2f, Fig. S2g-j). We then treated cells with zeocin and assayed RPA-foci as proxy of resection. We observed that also under these conditions, chemical inhibition of DDK using XL413 suppressed RPA-foci formation and resection indicating that DDK is required for resection also in a CDK-proficient-state.

In these experiments, we also tested the induction of gamma-H2AX phosphorylation by zeocin via western blot and observed that addition of XL413 did not affect gamma-H2AX phosphorylation (Fig. S2h-j). As such, we conclude that DNA damage induction is independent of the status of DDK.

3. In Figure 3a, it seems that when DDK and CDK is combined in the kinase assay the phosphor-incorporation is less as compared to the CDK only sample. Why is that? Better explain this somewhere.

We thank the reviewer for this observation. We have to admit that we do not fully understand where this apparent negative effect is coming from. As we added

two active kinases to the assay, it is entirely possible that there is trans-phosphorylation. Since currently, we do not understand the molecular basis of this observation and do neither know if it has resemblance to the *in vivo* situation, we have decided to take out these data from the current manuscript (new Fig. 3a).

Instead, we focus on the DDK-phosphorylation of the CDK-phosphomimetic Sae-S267E (Fig. 3c) and would like to point out that DDK-phosphorylation leads to an additional Sae2-dependent stimulation of Mre11 activity in this context. From this additivity we conclude that DDK- and CDK-phosphorylation affects Sae2 function by mechanisms that are (at least partially) independent.

Reviewer #2 (Remarks to the Author):

In this manuscript Galanti and co-workers describe the function of the CDC7/DBF4 kinase in the regulation of DNA double-strand breaks resection and the consequences of impaired CDC7/DBF4 activity on homologous recombination in budding yeast.

The authors combine molecular genetics, proteomics, biochemical and functional approaches, showing that two nucleases, Sae2/MRN and DNA2, are the target of CDC7/DBF4 activity, regulating short- and long-range resection respectively. They provide substantial mechanistic insight on how CDC7 works together with CDK1 in determining the cell cycle dependent pathway choice in dsDNA breaks repair. Finally, the authors also provide some evidence that the role of CDC7 in resection is conserved in human cells.

The experiments performed are well-designed, consequential, and elegant; the data are clearly presented and fully supports the claims. Methodology is rigorous and of very high standard throughout; the manuscript is well written and sufficient detail is given in the M&M section.

This is an important work clarifies how two kinases, CDC7/DBF4 and CDK1, which together have a major role in driving cells through the cycle-cycle, commonly regulate HR, and it is a significant advance in the field of genome stability.

We thank this reviewer for his/her considerations and the careful assessment of our work. We very much appreciate the highly positive feedback.

I only have a couple of minor observations:

1) the experiment which shows that CDC7 inhibition potentially affect resection in U2OS cells (Fig 2FG) is performed in cells also treated with a CDK1 inhibitor (RO-

3306), which is likely to also affect efficiency of resection. This should be mentioned and discussed.

We thank the reviewer for raising this important point. Indeed, we agree with the reviewer that CDK1 inhibition in the RO-3306 is likely to affect resection. To ensure that the observed effects also hold true in a state of high CDK1 activity, we have now replaced the previous experiment with CDK1 inhibition with a new line of experiments, where we arrested U2OS cells in metaphase and in a state of high CDK1 and CDK2 activity using nocodazole (shown in Fig. 2f, S2g-j). In these experiments, we then treated cells with zeocin and assayed RPA-foci as proxy of resection. We observed that also under these conditions, chemical inhibition of DDK using XL413 suppressed RPA foci formation and resection. This indicates that DDK is required for resection also in a CDK-proficient-state.

2) In Fig3 g-h is not clear if the differences in short-range resection between WT and *dbf4-3aid* are significant and if in these experimental conditions resection is defective or simply delayed. Can the effect of CDK1 inhibition be compared in the same assay?

We apologize for the omission of statistical data in the qPCR-based resection experiments. We have now included statistical data (two-sided t-tests) for Fig. 3g-h, Fig. 4e-f, Fig. S3 e-f). For the specific experiments of Fig. 3g-h, which assay short-range resection by the Mre11 complex, we find that resection in *dbf4-3aid* cells is significantly lower than in WT cells ($p=0.002$) in distance of 98 bp from DSB, but that at 120bp from the DSB this difference is of borderline significance ($p=0.056$).

We also wanted to compare the effects that DDK and CDK phosphorylation may have on resection by the Mre11 complex. Here, we employed the *sae2-S267A* mutant, which is defective in CDK phosphorylation of the major CDK site on Sae2. As can be seen in the new Figure S3E-G, we observed that Mre11-dependent resection was largely deficient in the absence of CDK-phosphorylation of Sae2 (*sae2-S267A*), while it was only partially inhibited in the absence of DDK (Fig. 3g-h S3e-g). Lastly, we directly compared the effect of mutating DDK phosphorylation sites on Sae2 (*sae2-6A*) to the effect of mutating the major CDK phosphorylation (*sae2-267A*) and to the combination mutant (*sae2-7A*). Consistent with the previous resection and sensitivity data, we found that the *sae2-267A* mutant strains showed greater CPT sensitivity than the *sae2-6A* mutant strain deficient in DDK phosphorylation of Sae2 (Fig. 3I). Notably, mutating both CDK and DDK phosphorylation sites in the *sae2-7A* strain led to a further increase in CPT sensitivity, suggesting that CDK and DDK phosphorylation regulate Sae2 and the Mre11 complex by at least partially independent mechanisms (Fig. 3I, Fig. S3I).

Reviewer #3 (Remarks to the Author):

The study reported by Galanti et al in the current manuscript being considered for publication in Nature Communications makes several important contributions to the DNA repair field. Chief among these are the findings that: (i) DDK promotes Homologous Recombination following DNA double-strand breaks, through stimulation of end resection, (ii) DDK appears to do so through phosphorylation of key factors involved in resection initiation (Sae2) and resection elongation (Dna2), (iii) engineered DDK activation in G1 phase can result in limited resection at a cell cycle stage where this does not normally occur.

Overall, the work is presented in a clear fashion, with convincing results that back up the main claims made by the authors. In my view, the main value to the field is in uncovering mechanistic details of budding yeast DDK promotion of HR-directed repair of DNA DSBs. While some initial results, and previous cited reports, are consistent with conservation of this role for DDK in humans, this is not explored in a substantive way in the present manuscript.

We appreciate the positive feedback by this reviewer and thank the reviewer for his/her considerations and the careful assessment of our work.

I am supportive of publication, but the following points should be considered for a revised manuscript:

1 - The evidence supporting conservation in human cells of a DDK role in promoting HR-directed DNA DSB repair is very limited, consisting of only the data in Fig. 2F. Was there confirmation of M-phase arrest of the U2OS cells treated with RO-3306? While flow cytometry analysis was performed to confirm cell-cycle arrest for yeast cells in several instances, I saw no mention of this being carried out for the U2OS cells. Given that RO-3306 is a CDK1 inhibitor, it is important to factor dual inhibition into the interpretation of the DDK impairment results. Was consideration given to inhibiting DDK through another means in the U2OS cells (e.g. siRNA-mediated depletion)?

We thank the reviewer for raising this important point. As pointed out by all three reviewers the use of RO-3306 to induce a G2 arrest comes with the caveat of CDK1 inhibition. As currently we do not know in how far CDK2 can compensate for the regulation of resection, we decided to replace the experiment with another line of experiments which used nocodazole-arrested cells, where CDK1 and CDK2 are fully active (shown in Fig. 2f, Fig. S2g-j). Specifically, we treated nocodazole-arrested cells with zeocin and assayed RPA-foci as proxy of resection. We observed that also under these conditions, chemical inhibition of DDK using

XL413 suppressed RPA foci formation and resection (Fig. 2f). This indicates that DDK is required for resection also in a CDK-proficient-state and not only in a G2 arrest induced by CDK1-inhibition. In these experiments we also verified the quality of the cell cycle arrest by flow cytometry as suggested by the reviewer (Fig. S2g). Finally, we would like to note that under the experimental conditions used in our experiments, we obtained partial inhibition of DDK (Fig. S2h-i)). We agree that this could be further improved by depletion experiments (for example siRNA), but caution that the prominent role of DDK in the cell cycle will make experiments with prolonged times of DDK depletion as typical in an siRNA experiment very difficult.

2 - For the *sae2-14A* strain, the lack of growth on even low CPT concentrations (Fig.3K) may also have been due to lower stability of the corresponding mutant Sae2 protein, similarly to what was noted by the authors in relation to the defect in its phosphorylation

We agree that the *sae2-14A* has a similar CPT phenotype as the *sae2Δ* mutant. This finding together with the expression defect suggests that the mutant protein is not functional and potentially is misfolded. Thus, while some of the mutated phosphorylation sites may be DDK target sites, we have not considered this mutant for further analysis (i.e. epistasis analysis with CDK sites), moved the according experiments to supplementary data (Fig. S3j-k) and have instead focused on the *sae2-6A* in the main manuscript.

3 - The authors indicate for their assessment of recombination-mediated repair (Fig. 5D), that cells were maintained in a state of G1 arrest throughout the experiment, however the flow cytometry profiles (Fig.S5E) seem to indicate some progression into S-phase for all the strains examined for the 5-9h timepoints

We thank the reviewer for this observation. In long G1 arrests, we (and others) typically observe a slight shift in the FACS signal (see Reusswig et al. Nat Commun 2022; Seel et al. Nat Struct Mol Biol 2023). This shift is due to continuous cell growth and/or mitochondrial DNA synthesis, but does not reflect entry into S-phase. To ensure that in the experiments of Fig. 5 there is no leakage into S-phase we determined the budding index (new Fig. S5i), which shows that there is no increase in budded cells over time course of the experiment.

4 - Typos: Last paragraph of Results, instead of “We ruled out that these effects...”, seems to me it should be “We confirmed that these effects...”; there are a couple of instances where the phrase “difficult to entangle” appears when it should be “difficult to disentangle”

Corrected.

REVIEWERS' COMMENTS

Reviewer #1 (Remarks to the Author):

The authors have clearly answered to the request.

Reviewer #2 (Remarks to the Author):

The authors have satisfactory addressed my comments.

Reviewer #3 (Remarks to the Author):

I would like to congratulate the authors on the revised version of their manuscript. The modifications have satisfactorily addressed the main concerns I had expressed. There remains one typo, "entangle" should be replaced with "disentangle" on p.18

Point-by-point response to reviewer comments to revised version of Galanti et al. "Dbf4-dependent kinase promotes cell-cycle controlled resection of DNA double-strand breaks and repair by homologous recombination"

Reviewer #1 (Remarks to the Author):

The authors have clearly answered to the request.

We thank the reviewer for his/her considerations and the careful assessment of our work. We appreciate the effort put into peer-review.

Reviewer #2 (Remarks to the Author):

The authors have satisfactorily addressed my comments.

We thank the reviewer for his/her considerations and the careful assessment of our work. We appreciate the effort put into peer-review.

Reviewer #3 (Remarks to the Author):

I would like to congratulate the authors on the revised version of their manuscript. The modifications have satisfactorily addressed the main concerns I had expressed. There remains one typo, "entangle" should be replaced with "disentangle" on p.18

We thank the reviewer for his/her considerations and the careful assessment of our work. We appreciate the highly positive feedback and the effort put into peer-review. We corrected the remaining typo.